# A simplified atmospheric boundary layer model for an improved representation of air-sea interactions in eddying oceanic models: implementation and first evaluation in $\mathrm{NEMO(4.0)}$

F. Lemarié[1], G. Samson[2], J.L. Redelsperger[3], H. Giordani[4], T. Brivoal[2], and G. Madec[5,1]

[1]Univ. Grenoble Alpes, Inria, CNRS, Grenoble INP, LJK, 38000 Grenoble, France
[2]Mercator Océan, Toulouse, France
[3]Univ. Brest, CNRS, IRD, Ifremer, Laboratoire d'Océanographie Physique et Spatiale (LOPS), IUEM, Brest, France
[4]Centre National de Recherches Météorologiques (CNRM), Université de Toulouse, Météo-France, CNRS, Toulouse, France
[5]Sorbonne Universités (UPMC, Univ Paris 06)-CNRS-IRD-MNHN, LOCEAN Laboratory, Paris, France

**Correspondence:** Florian Lemarié (florian.lemarie@inria.fr)

**Abstract.** A simplified model of the Atmospheric Boundary Layer (ABL) of intermediate complexity between a bulk parameterization and a three-dimensional atmospheric model is developed and integrated to the Nucleus for European Modelling of the Ocean (NEMO) general circulation model. An objective in the derivation of such simplified model called ABL1d is to reach an apt representation in ocean-only numerical simulations of some of the key processes associated to air/sea interactions at the characteristic scales of the oceanic mesoscale. In this paper we describe the formulation of the ABL1d model and the strategy to constrain this model with large-scale atmospheric data available from reanalysis or real-time forecasts. A particular emphasis is on the appropriate choice and calibration of a turbulent closure scheme for the atmospheric boundary layer. This is a key ingredient to properly represent the air/sea interaction processes of interest. We also provide a detailed description of the NEMO-ABL1d coupling infrastructure and its computational efficiency. The resulting simplified model is then tested for several boundary-layer regimes relevant to either ocean/atmosphere or sea-ice/atmosphere coupling. The coupled system is also tested with a realistic $0.25°$ resolution global configuration. The numerical results are evaluated using standard metrics from the literature to quantify the wind/sea surface temperature (a.k.a. thermal feedback effect), wind/currents (a.k.a. current feedback effect) and ABL/sea-ice couplings. With respect to these metrics, our results show very good agreement with observations and fully coupled ocean-atmosphere models for a computational overhead of about $9\%$ in term of elapsed time compared to standard uncoupled simulations. This moderate overhead, largely due to I/O operations, leaves room for further improvement to relax the assumption of horizontal homogeneity behind ABL1d and thus to further improve the realism of the coupling while keeping the flexibility of ocean-only modelling.

## 1 Introduction

Owing to advances in computational power, global oceanic models used for research or operational purposes are now configured with increasingly higher horizontal/vertical resolution thus resolving the baroclinic deformation radius in the tropics (e.g. Deshayes et al., 2013; Metzger et al., 2014; von Schuckmann et al., 2018). Meanwhile fine-scale local models are routinely

used to simulate submeso-scales, which occur on scales on the order of $0.1 - 20$ km horizontally, and their impact on larger scales (e.g. Marchesiello et al., 2011; McWilliams et al., 2019). By increasing the oceanic model resolution, small scales features are explicitly resolved but an apt representation of the associated processes requires the relevant scales to be present also in the surface forcings including the proper interaction with the low-level atmosphere.

## 1.1 Historical context

Historically, Oceanic General Circulation Models (OGCMs) were forced by specified wind-stress and thermal boundary condition (from observations or reanalysis) independent from the oceanic state thus often leading to important drifts in model sea surface properties. To minimize such drifts, a flux correction in the form of a restoring of sea-surface temperature and salinity toward climatological values can be added (e.g. Haney, 1971; Barnier et al., 1995). To overcome the shortcomings of the forcing with specified flux, Takano et al. (1973) proposed to use a parameterization of the atmospheric surface layer (ASL) constrained by large scale meteorological data and by the sea-state (essentially the sea-surface temperature and sometimes the surface currents) to compute the turbulent components of air-sea fluxes. Currently, whatever the target applications, such technique is widely used in the absence of a concurrently running atmospheric model. Such parameterization of the ASL (known as bulk parameterization, e.g. Beljaars, 1995; Large, 2006), which corresponds to a generalization of the classical neutral wall law to stratified conditions (Monin and Obukhov, 1954), is expected to be valid in the first tenth of meters in the atmosphere. In practice, unless a fully coupled ocean-atmosphere model is used, atmospheric quantities at 10 meters, either from existing numerical simulations of the atmosphere or from observations, are prescribed as input to the bulk parameterization. Throughout the paper, this approach will be referred to as "ASL forcing strategy". A problem with such methodology is that the fast component of the system (the atmosphere) is specified to force the slow component (the ocean) whereas the inertia is in the latter. Indeed, a change in wind-stress or heat flux will affect 10 meter winds and temperature more strongly than sea surface currents and temperature. In the "ASL forcing strategy", the key marine atmospheric boundary layer (MABL) processes are not taken into account and thus feedback loops between the MABL and the upper-ocean are not represented.

## 1.2 Air-sea interactions at oceanic mesoscales

An increasing number of studies based either on observational studies and/or on air-sea coupled simulations have unambiguously shown the existence of air-sea interactions at oceanic mesoscales (e.g. Giordani et al., 1998; Bourras et al., 2004; Chelton and Xie, 2010; Frenger et al., 2013; Schneider and Qiu, 2015; Oerder et al., 2016). Those interactions affect the mass, heat and momentum exchange between the atmosphere and the ocean. We focus in this work on the dynamical response of the surface wind-stress to the sea-surface properties (Sea Surface Temperature (SST) and currents) which directly affects low-level winds, temperature and humidity. Several mechanisms responsible for the surface wind-stress response to SST and oceanic currents can be invoked:

(*i*) *Downward momentum mixing*: SST-induced changes in the stratification produce significant changes of wind speed and turbulent fluxes throughout the MABL with an increase (resp. decrease) in wind speed over warm water (resp. cold water). As the wind blows over warm water, the MABL becomes more unstable which leads to an increased vertical mixing resulting in a

downward mixing of momentum from the upper atmosphere to the surface strengthening surface winds on the warm side of an
      SST front (e.g. Wallace et al., 1989). This mechanism results in a proportional relationship between wind-stress intensity and
      SST mesoscale anomalies which has been identified in observations and coupled simulations (e.g. O'Neill et al., 2010; Oerder
      et al., 2016). Considering spatial derivatives of this proportional relationship leads to a wind-stress divergence (resp. curl) to
      downwind (resp. crosswind) SST gradients correlation (e.g. Chelton and Xie, 2010; Schneider and Qiu, 2015).

(*ii*) *Atmospheric pressure adjustment*: this mechanism corresponds to an adjustment of the atmospheric pressure gradient to
      the underlying SST which manifests itself as a linear relation between horizontal wind divergence and the Laplacian of SST
      (Lindzen and Nigam, 1987; Minobe et al., 2008; Lambaerts et al., 2013).

      (*iii*) *Oceanic current feedback*: The momentum exchange between the ocean and the atmosphere is also largely affected by
      a dynamical coupling through the dependence of surface wind-stress on oceanic surface currents (e.g. Dewar and Flierl, 1987).
This coupling results in a drag exerted by the air-sea interface on the ocean which leads to a systematic reduction of the wind
      power input to the oceanic circulation.

      Even if these three mechanisms are mainly active at oceanic eddy scales, they can induce significant effects at larger scales
      in regions with large SST gradients and/or surface currents (Hogg et al., 2009; Bryan et al., 2010; Renault et al., 2016a). They
      jointly leave their imprint on the wind divergence and identifying the relative importance of each mechanism on the momentum
balance is difficult because it depends on the dynamical regime and on the spatial and temporal scales of interest (Schneider
      and Qiu, 2015; Ayet and Redelsperger, 2019).

      In the ASL coupling strategy the pressure adjustment mechanism is absent and only a small fraction of the downward
      momentum mixing mechanism is accounted for through the modification of surface drag coefficient depending on the ASL
      stability (Businger and Shaw, 1984; Chelton and Xie, 2010). As far as the current feedback is concerned, Renault et al. (2016b)
showed that the reduction of wind power input to the ocean is systematically over-estimated in oceanic simulations based on
      an ASL forcing strategy compared to air-sea coupled simulations. A simulation that neglects the MABL adjustment to the
      current feedback can not represent the partial re-energization of the ocean by the atmosphere and hence overestimates the drag
      effect by more than $30\%$ (e.g. Renault et al., 2016b, 2019a). The ASL forcing strategy used in most oceanic models will thus
      over-estimate the current feedback effect and under-estimate the downward momentum mixing.

## 1.3  The proposed approach and focus for this paper

      The various aspects discussed so far suggest that a relevant coupling at the characteristic scales of the oceanic mesoscales
      requires nearly the same horizontal resolution in the ocean and the atmosphere (since the atmosphere must "see" oceanic
      eddies and fronts) as well as an atmospheric component more complete than a simple ASL parameterization to estimate air-sea
      fluxes. This assessment raises numerous questions on current practices to force oceanic models across all scales[1] in the absence
of an interactive atmospheric model. The computational cost associated to the systematic use of fully coupled ocean atmosphere
      models of similar horizontal resolution is generally unaffordable and comes with practical issues like the proper definition of

      ---

      [1]This remark is supported by the conclusions of the CLIVAR Working Group on Model Development following the Kiel meeting in Apr. 2014 http://www.clivar.org/sites/default/files/documents/exchanges65_0.pdf

initial conditions via data assimilation techniques (e.g. Mulholland et al., 2015) and the proper choice of a parameterization set. Moreover, in the fully coupled case at basin or global interannual scales the temporal consistency with the observed variability is generally lost unless a nudging toward observations or reanalysis is done in the atmosphere above the MABL (e.g. Bielli et al., 2009).

There is thus clearly room for improvement in the methodology to compute the surface boundary conditions for an ocean model. Alternatives to the ASL forcing strategy have already been suggested by Kleeman and Power (1995); Seager et al. (1995) and Deremble et al. (2013). They proposed a vertically integrated thermodynamically active and dynamically passive MABL model where the wind and the MABL height are specified as in the current practices. Such model allows a better feedback between SSTs and low-level air temperature and humidity because the latter are prognostic (Abel, 2018). However, by construction, such model do not reproduce the various aforementioned coupling mechanisms affecting the surface wind stress. Their focus is on the improvement of the large-scale thermodynamics while ours is on the improvement of the eddy-scale momentum exchanges.

In the present study we propose an alternative methodology to improve the representation of the downward momentum mixing and of the current feedback effect in ocean-only simulations, leaving aside the pressure gradient adjustment for now on. Our aim is to account for the modulation of atmospheric turbulence by anomalies in sea-surface properties in the air-sea fluxes computation which is thought to be the main coupling mechanism at the characteristic scales of the oceanic mesoscales. As a step forward beyond the ASL forcing strategy we propose to complement the ASL parameterization with an ABL parameterization while keeping a single-column frame. By construction our approach excludes horizontal advection whose effect can be important at the vicinity of strong SST fronts (e.g. Kilpatrick et al., 2014; Ayet and Redelsperger, 2019). However we considered that finding a simple and efficient MABL parameterization is the top priority to start investigating the viability of our approach in terms of practical implementation and computational cost. Indeed there exists a large variety of parameterization schemes to represent the effects of subgrid-scale turbulent mixing in the ABL (see LeMone et al., 2019, and references therein). The schemes based on a diagnostic or prognostic turbulent kinetic energy (TKE) are very popular for operational and research purposes despite well-identified shortcomings (e.g. Baklanov et al., 2011). For our purposes we do not need the full complexity of the schemes used in practice in atmospheric models because aspects like cloud processes and complex terrains are outside our scope. For this reason, the guideline in this paper is the development and the testing of a simplified version of the TKE-based scheme proposed by Cuxart et al. (2000) for over-water and over-sea-ice conditions. Note that the single-column approximation for our simplified model selected in this study is only a temporary choice to provide evidence on the viability of the whole approach. More advanced formulations allowing to recover a more realistic momentum balance (i.e. including advection) will be studied in future work.

## 1.4 Content

The objective of the present study is to introduce a simplified model of the MABL of intermediate complexity between a bulk parameterization and a full three-dimensional atmospheric model and to describe its integration to the Nucleus for European Modelling of the Ocean (NEMO) general circulation model (Madec, 2012). This approach will be referred to as "ABL coupling

strategy". A constraint in the conception of such a simplified model is to allow an apt representation of the downward momentum mixing mechanism and partial re-energization of the ocean by the atmosphere while keeping the computational efficiency and flexibility inherent to ocean only modeling. The paper is organized as follows. In Sect. 2, we describe the continuous formulation of the simplified model called ABL1d, including the parameterization scheme used to represent vertical turbulent mixing in the MABL and the strategy to constrain this model with large-scale atmospheric conditions. Sect. 3 provides the description of the dicretization and of the practical implementation of the ABL1d model in the NEMO framework. In Sect. 4 and 5 numerical results obtained for some atmosphere-only simplified test-cases available in the literature and for a coupled NEMO- ABL1d simulation in a global configuration are shown. Finally, our conclusions and perspectives are summarized and discussed in Sect. 6.

## 2 Model equations

In this section we first provide some basic elements on model reduction to motivate our approach and mention possible alternatives (Subsect. 2.1). Then we detail the continuous formulation of the ABL1d model and discuss the assumptions made. In particular the governing equations and necessary boundary conditions are given in Subsect. 2.2 and the turbulence closure scheme for the MABL in Subsect. 2.3. Finally in Subsect. 2.4 we discuss the methodology to relax the ABL1d prognostic quantities toward large-scale data.

### 2.1 Motivations and proposed approach

Global oceanic models can be run at higher resolution that global atmospheric models because of their affordable computational cost. From an oceanic perspective, we generally simulate at high-resolution (in space and time) ocean fields $\phi_{\text{HR}}^{\text{oce}}(x,y,z,t)$ over a time interval $t \in [0,T]$ over which only large scale atmospheric data $\phi_{\text{LS}}^{\text{atm}}(x,y,z,t)$ are known from the integration of a model $\mathcal{M}_{\text{atm}}$ using lower-resolution surface oceanic data $\phi_{\text{LS}}^{\text{oce}}(x,y,z=0,t)$ to compute its surface boundary conditions, namely

$$\phi_{\text{LS}}^{\text{atm}}(x,y,z,t) = \mathcal{M}_{\text{atm}}(\phi_{\text{LS}}^{\text{oce}}(x,y,z=0,t)), \qquad t \in [0,T]$$

Instead of directly using $\phi_{\text{LS}}^{\text{atm}}(x,y,z=10\,\text{m},t)$ to constrain the oceanic model as in the ASL forcing strategy, our objective is to estimate (without running the full atmospheric model again) the correction to the $10\,\text{m}$ large-scale atmospheric data associated both with the fine resolution in the oceanic surface fields and with the two-way air-sea coupling. Somehow we aim at finding a methodology to get a cheap estimate $\widetilde{\phi}_{\text{HR}}^{\text{atm}}(x,y,z=10\,\text{m},t)$ of the solution that would have been obtained using a coupling of $\mathcal{M}_{\text{atm}}$ and the oceanic model at high resolution. To do so we could imagine several approaches: *(i)* estimate $\dfrac{\partial \mathcal{M}_{\text{atm}}}{\partial \phi_{\text{LS}}^{\text{oce}}}$ (i.e. the derivatives of the atmospheric solution with respect to the oceanic parameters) via sensitivity analysis which would require to have the possibility to operate $\mathcal{M}_{\text{atm}}$; *(ii)* Build a surrogate model via learning strategies which would require a huge amount of data and computing time; *(iii)* Select the feedback loops of interest and define a simplified model to mimic the underlying physical mechanisms. Following the terminology of Razavi et al. (2012), the first two approaches enter the class of statistical or empirical data-driven models emulating the original model responses while the third one enters the class

of low-fidelity physically-based surrogates which are built on a simplified version of the original system of equations. In the present study we consider this latter approach, in the spirit of Giordani et al. (2005) who derived a simplified oceanic model by degenerating the primitive equations system and prescribing geostrophic currents into the momentum equation in substitution of the horizontal pressure gradient. In this model, a simple 1D oceanic mixed layer is three-dimensionalized via advective terms to couple the vertical columns with each other. The idea here is to translate this idea to the MABL context. In the rest of this section we describe the continuous formulation of our simplified MABL model which will be referred to as ABL1d.

## 2.2 Formulation of a single column approach

The formulation of the ABL1d model is derived under the following assumptions: (*i*) horizontal homogeneity (i.e. $\partial_x \cdot = \partial_y \cdot = 0$), (*ii*) the atmosphere in the computational domain is transparent (i.e. $\partial_z \mathcal{I} = 0$ with $\mathcal{I}$ the radiative flux) meaning that cloud physics is ignored and solar radiation and precipitations at the air-sea interface are specified as usual from observations (e.g. Large and Yeager, 2009) (*iii*) vertical advection is neglected. Such assumptions prevent the model to prognostically account for the SST-induced adjustment of the atmospheric horizontal pressure gradient and for horizontal advective processes associated with a higher resolution boundary condition at the air-sea interface. The focus here is on the proper representation of the modulation of the MABL turbulent mixing by the air-sea feedback which is thought to be the main coupling mechanism at the characteristic scales of the oceanic mesoscales impacting $\phi^{\mathrm{atm}}(z = 10\,\mathrm{m}, t)$ hence air-sea fluxes. This mechanism is expected to explain most of the eddy-scale wind-SST and wind-currents interactions and is key to properly downscale large-scale atmospheric data produced by a coarse resolution GCM to the oceanic resolution.

At a given location in space, the ABL1d model for the Reynolds-averaged profiles of horizontal velocities $\mathbf{u}_h(z, t)$, potential temperature $\theta(z, t)$, and specific humidity $q(z, t)$, provided a suitable initial condition, is

$$
\begin{cases}
\partial_t \mathbf{u}_h &= -f \mathbf{k} \times \mathbf{u}_h + \partial_z \left( K_m \partial_z \mathbf{u}_h \right) + \mathbf{R}_{\mathrm{LS}} \\
\partial_t \theta &= \partial_z \left( K_s \partial_z \theta \right) + \lambda_s (\theta_{\mathrm{LS}} - \theta) \\
\partial_t q &= \partial_z \left( K_s \partial_z q \right) + \lambda_s (q_{\mathrm{LS}} - q)
\end{cases}
\tag{1}
$$

for the height $z$ between a lower boundary $z_{\mathrm{sfc}}$ and an upper boundary $z_{\mathrm{top}}$ which will be considered horizontally constant because only the ocean and sea-ice covered areas are of interest. In (1), $\mathbf{k} = (0, 0, 1)^t$ is a vertical unit vector, $f$ the Coriolis parameter, $K_m$ and $K_s$ are the eddy diffusivity respectively for momentum and scalars, the subscript LS is used to characterize large-scale quantities known a priori, $\lambda_s(z, t)$ is the inverse of a relaxation timescale, and $\mathbf{R}_{\mathrm{LS}}$ denotes a large-scale forcing for the momentum equation. $\mathbf{R}_{\mathrm{LS}}$ can either represent a forcing by geostrophic winds $\mathbf{u}_G$ (i.e. $\mathbf{R}_{\mathrm{LS}} = f \mathbf{k} \times \mathbf{u}_G$) or equivalently by a horizontal pressure gradient (i.e. $\mathbf{R}_{\mathrm{LS}} = \left( \frac{1}{\rho_a} \boldsymbol{\nabla}_h p \right)_{\mathrm{LS}}$) combined with a standard Newtonian relaxation (i.e. $\mathbf{R}_{\mathrm{LS}} = \lambda_m (\mathbf{u}_{\mathrm{LS}} - \mathbf{u}_h)$). Because of the simplifications made to derive the ABL1d model the $\mathbf{R}_{\mathrm{LS}}$ term and a nonzero $\lambda_s$ are necessary to prevent the prognostic variables to drift very far away from the large-scale values used to "guide" the model. By itself a relaxation term does not represent directly any real physical process, but the rationale is that it accounts for the influence of large-scale three-dimensional circulation processes not explicitly represented in a simple 1d model. Note that this methodology is currently used to evaluate GCM parameterizations in 1D column model. Once the turbulent mixing and the Coriolis term have been computed

to provide a provisional prediction $\phi^{n+1,\star}$ at time $n+1$ for any ABL1d prognostic variable $\phi$, the relaxation term provides a weighting between this prediction and the large-scale quantities

$$\phi^{n+1} = \Delta t \lambda \, \phi_{\text{LS}} + (1 - \Delta t \lambda)\phi^{n+1,\star} \tag{2}$$

with $\Delta t$ the increment of the temporal discretization. Above the boundary layer, the ABL1d formulation is unable to properly represent the physics, therefore the $\lambda$ parameter should be large while in the first tens of meters near the surface we expect the ABL1d model to accurately represent the interaction with the fine resolution oceanic state and thus the relaxation toward $\phi_{\text{LS}}$ should be small. The exact form of the $\lambda_s$ and $\lambda_m$ coefficients is discussed later in Sec. 2.4. Note that because of the relaxation term, three-dimensional atmospheric data for $\mathbf{u}_{\text{LS}}$, $\theta_{\text{LS}}$, $q_{\text{LS}}$, and possibly $\left( \frac{1}{\rho_a} \boldsymbol{\nabla}_h p \right)_{\text{LS}}$ sampled between $z_{\text{sfc}}$ and $z_{\text{top}}$ must be provided to the oceanic model instead of the two-dimensional data (usually at $10\,\text{m}$) necessary for an ASL forcing strategy. Since the ABL1d model does not include any representation of radiative processes and microphysics, the radiative fluxes and precipitation at the air-sea interface are similar to the one provided for a standard uncoupled oceanic simulation. The model requires boundary conditions for the vertical mixing terms which are computed via a standard bulk formulation:

$$K_m \partial_z \mathbf{u}_h|_{z=z_{\text{sfc}}} = C_D \|\mathbf{u}_h(z_{\text{sfc}}) - \mathbf{u}_{\text{oce}}\|(\mathbf{u}_h(z_{\text{sfc}}) - \mathbf{u}_{\text{oce}}), \tag{3}$$

$$K_s \partial_z \phi|_{z=z_{\text{sfc}}} = C_\phi \|\mathbf{u}_h(z_{\text{sfc}}) - \mathbf{u}_{\text{oce}}\|(\phi(z_{\text{sfc}}) - \phi_{\text{oce}}), \qquad \text{with } \phi = \theta, q \tag{4}$$

For the sake of consistency, it is preferable to use a bulk formulation as close as possible to the one used to compute the three-dimensional large-scale atmospheric data $\phi_{\text{LS}}^{\text{atm}}$. Because in the present study the plan is to use a large-scale forcing from ECMWF reanalysis products, we use the IFS[2] bulk formulation such as implemented in the AeroBulk[3] package (Brodeau et al., 2017) to compute $C_D$, $C_\theta$, and $C_q$ in realistic simulations (see Sect. 5). Note that for an ASL forcing strategy $\mathbf{u}_h(z_{\text{sfc}})$ and $\phi(z_{\text{sfc}})$ in (3) would be respectively equal to $\mathbf{u}_{\text{LS}}(z = 10\,\text{m})$ and $\phi_{\text{LS}}(z = 10\,\text{m})$ while in the ABL coupling strategy those variables are provided prognostically by an ABL1d model. As far as the boundary conditions at $z = z_{\text{top}}$ are concerned, Dirichlet boundary conditions $\mathbf{u}_h(z_{\text{top}}) = \mathbf{u}_{\text{LS}}(z_{\text{top}})$ and $\phi(z_{\text{top}}) = \phi_{\text{LS}}(z_{\text{top}})$ are prescribed.

Model (1) is a first step before evolving toward a more advanced surrogate model including horizontal advection and fine-scale pressure gradient in the future. A particular focus of the present study is on the appropriate choice of a closure scheme to diagnose the eddy diffusivities $K_m$ and $K_s$. This is a key step to properly represent the downward mixing process.

### 2.3 Turbulence closure scheme

This subsection describes the turbulence scheme used to compute the eddy diffusivity for momentum and scalars. Those eddy diffusivities are responsible for a vertical mixing of atmospheric variables due to turbulent processes. The turbulence scheme we have implemented in our ABL1d model is very similar to the so-called CBR-1d scheme of Cuxart et al. (2000) which is used operationally at Meteo France (Bazile et al., 2012). We chose to recode the parameterization from scratch for several reasons: computational efficiency, consistency with the NEMO coding rules, use of a geopotential vertical coordinate, and flexibility to add elements specific of the marine atmospheric boundary layer.

---

[2]Integrated Forecasting System: https://www.ecmwf.int/en/forecasts/documentation-and-support/changes-ecmwf-model/ifs-documentation

[3]http://aerobulk.sourceforge.net/

CBR-1d is a one-equation turbulence closure model based on a prognostic turbulent kinetic energy (TKE) and a diagnostic computation of appropriate length scales. The prognostic equation for the TKE $e = \frac{1}{2}\left(\langle u'u'\rangle + \langle v'v'\rangle + \langle w'w'\rangle\right)$ (with $\langle \cdot \rangle$ the Reynolds averaging operator) is

$$\partial_t e = -\langle \mathbf{u}_h'w'\rangle \cdot \partial_z \langle \mathbf{u}_h\rangle + \frac{g}{\theta_v^{\mathrm{ref}}}\langle w'\theta_v'\rangle - \partial_z\left(\langle e'w'\rangle + \frac{1}{\rho_a}\langle p'w'\rangle\right) - \varepsilon \tag{5}$$

where horizontal terms and vertical advection are neglected, as usually done in mesoscale atmospheric models. Here $\theta_v$ is the virtual potential temperature, $\rho_a$ is atmospheric density, and $\varepsilon$ a dissipation term. In order to express the evolution of $e$ in terms of Reynolds averaged atmospheric variables we consider the standard closure assumptions for the first order turbulent fluxes (Cuxart et al., 2000) to obtain the classical TKE prognostic equation

$$\partial_t e = K_m \|\partial_z \langle \mathbf{u}_h\rangle\|^2 - K_s N^2 + \partial_z (K_e \partial_z e) - \frac{c_\varepsilon}{l_\varepsilon} e^{3/2} \tag{6}$$

where $l_\varepsilon$ is a dissipative length scale, $c_\epsilon$ a constant, and $N^2$ is the moist Brunt-Väisälä frequency computed as $N^2 = (g/\theta_v^{\mathrm{ref}})\left(\partial_z \langle\theta\rangle + 0.608\,\partial_z(\langle\theta\rangle\langle q\rangle)\right)$ with $\theta_v^{\mathrm{ref}} = 288$ K. The eddy diffusivities for momentum $K_m$, TKE $K_e$ and scalars $K_s$ all depend on $e$ and on a mixing length scale $l_m$

$$(K_m, K_s, K_e) = (C_m, C_s\phi_z, C_e)l_m\sqrt{e}$$

with $(C_m, C_s, C_e)$ a triplet of constants and $\phi_z$ a stability function proportional to the inverse of a turbulent Prandtl number, given by $\phi_z(z) = \left(1 + \max\left\{C_1 l_m l_\varepsilon N^2/e, -0.5455\right\}\right)^{-1}$. The $\phi_z$ function is bounded not to exceed $\phi_z^{\mathrm{max}} = 2.2$ as done in the Arpege model of Meteo-France (e.g. Bazile et al., 2012). Assuming that the minimum of $\phi_z$ is attained in the linearly stratified limit (i.e. for $l_m = l_\varepsilon = \sqrt{2e/N^2}$), values of the maximum Prandtl number $\mathrm{Pr}_t = C_m/(C_s\phi_z)$ are given in Tab. 1. Constant values for $C_m$, $C_s$, $C_e$, $c_\varepsilon$, and $C_1$ can be determined from different methods, leading to nearly similar values. The traditional way is to use the inertial-convective subrange theory of locally isotropic turbulence (Lilly, 1967; Deardorff, 1974). Another way relies on a theoretical turbulence model partly based on renormalization group methods (see Cheng et al., 2002). For the present study, the sets proposed by Cuxart et al. (2000) and Cheng et al. (2002) will be considered (Tab. 1). A major difference between the two sets concerns the value of $C_m$. This difference is explained by a reevaluation of the energy redistribution among velocity components by pressure fluctuations, whose magnitude is assumed to be proportional to the degree of energy anisotropy as initially introduced by Rotta (1951). Note that the constant set of Cheng et al. (2002) is now used by default in both research and operational Meteo-France models.

The Dirichlet boundary condition for TKE applied at the top $z = z_{\mathrm{top}}$ is $e(z = z_{\mathrm{top}}) = e_{\min} = 10^{-6}$ m$^2$ s$^{-2}$ and at the bottom $z = z_{\mathrm{sfc}}$ we have

$$e(z = z_{\mathrm{sfc}}) = e_{\mathrm{sfc}} = \frac{u_\star^2}{\sqrt{C_m c_\varepsilon}} + 0.2w_\star^2 \tag{7}$$

with $u_\star$ and $w_\star$ the friction and convective velocities given by the bulk formulation. The value for $e_{\min}$ has been chosen empirically as well as background values $K_m^{\min} = 10^{-4}$ m$^2$ s$^{-1}$ and $K_s^{\min} = 10^{-5}$ m$^2$ s$^{-1}$ for eddy diffusivities. The minimum

| | $C_m$ | $C_s$ | $C_e$ | $c_\varepsilon$ | $C_1$ | $\mathrm{Pr}_t^{\min}$ | $\mathrm{Pr}_t^{\max}$ | $l_m^{\min}$ |
|---|---|---|---|---|---|---|---|---|
| Cuxart et al. (2000) (CBR00) | 0.0667 | 0.1667 | 0.4 | 0.7 | 0.139 | 0.182 | 0.511 | 1.5m |
| Cheng et al. (2002) (CCH02) | 0.126 | 0.143 | 0.34 | 0.845 | 0.143 | 0.182 | 0.515 | 0.79m |

**Table 1.** Set of turbulence scheme constants from Cuxart et al. (2000) and Cheng et al. (2002). $\mathrm{Pr}_t = C_m/(C_s\phi_z)$ is the turbulent Prandtl number.

value for $l_m$ is simply set as $l_m^{\min} = \frac{K_m^{\min}}{C_m\sqrt{e_{\min}}}$. There are multiple options to compute the mixing lengths $l_m$ and $l_\varepsilon$ (this point will be discussed later in Sec. 3.2.2) but all options have identical boundary conditions $l_m(z = z_{\mathrm{top}}) = l_m^{\min}$ and

$$l_m(z = z_{\mathrm{sfc}}) = L_{\mathrm{sfc}} = \kappa \frac{(C_m c_\varepsilon)^{1/4}}{C_m}(z_{\mathrm{sfc}} + z_0) \tag{8}$$

Again the value of $L_{\mathrm{sfc}}$ results from the similarity theory in the neutrally stratified surface layer (Sec. 4.1 in Redelsperger et al., 2001, and App. A). In (8), $\kappa$ is the von Karman constant and $z_0$ a roughness length computed within the bulk algorithm. The way $e_{\mathrm{sfc}}$ and $L_{\mathrm{sfc}}$ are obtained is detailed in App. A

Our current implementation of boundary layer subgrid processes is an eddy-diffusivity approach which does not include any explicit representation of boundary-layer convective structures. This could be done via a mass-flux representation (e.g. Hourdin et al., 2002; Soares et al., 2004) or the introduction of a countergradient term (e.g. Troen and Mahrt, 1986). This point is left for future developments of the ABL1d model.

## 2.4 Processing of large-scale forcing and Newtonian relaxation

As mentioned earlier, the ABL1d model (1) requires three-dimensional $(x, y, z)$ large-scale atmospheric variables $\phi_{\mathrm{LS}}^{\mathrm{atm}}$ while existing uncoupled oceanic forcing strategies require only two-dimensional $(x, y)$ atmospheric variables. This is a difficulty for efficiency reasons since it substantially increases the number of I/Os but also for practical reasons because it requires the development of a dedicated tool to extract large-scale atmospheric data and interpolate them on prescribed geopotential heights from their native vertical grid which can be either pressure based or arbitrary Lagrangian Eulerian. Such tools have been developed specifically to work with ERA-Interim, ERA5 and operational IFS datasets and are described in App. B.

Beyond the particular values of $\phi_{\mathrm{LS}}^{\mathrm{atm}}$, the form of the relaxation timescale has great impact on model solutions. The vertical profile for the $\lambda_m$ and $\lambda_s$ coefficients in (1) is chosen to nudge strongly above the MABL and moderately in the MABL with a smooth transition between its minimum and maximum value to avoid large vertical gradients in $\lambda_m$ and $\lambda_s$ which would result in artificially large vertical gradients in atmospheric variables. In practice the $\lambda_m(z)$ and $\lambda_s(z)$ functions depend on the following parameters

- $(\lambda_m^{\max}, \lambda_m^{\min})$ and $(\lambda_s^{\max}, \lambda_s^{\min})$ which define the maximum and minimum of the nudging coefficient respectively for momentum and scalars. Following equation (2), a guideline to set reasonable values for those parameter values would be to make sure that $\Delta t \lambda_s^{\max} \approx 1$ (i.e. the large scale value is imposed above the boundary layer) and choose $\lambda_s^{\min}$ based

on the typical adjustment timescale of the ABL to surface perturbations. Broadly speaking the ABL can be defined as the region that respond to surface forcings with a timescale of about an hour (e.g. LeMone et al., 2019). In the realistic numerical experiments shown in Sect. 5, we used $\lambda_s^{\min} = \frac{1}{90[\min]}$ which, for an oceanic dynamical time-step $\Delta t = 1080$ s, would lead to $\Delta t \lambda_s^{\min} = 0.2$. (i.e. the boundary layer values are the result of a weighting with a weight $0.8$ for the ABL1d prediction and $0.2$ for the large scale value).

  – $(\beta_{\min}, \beta_{\max})$ which define the extent of the transition zone separating the maximum and minimum of the nudging coefficient

We considered the following general form for $\lambda_s(z)$ (resp. $\lambda_m(z)$), with $h_{bl}$ the boundary layer height whose value is diagnosed using an integral Richardson number criteria (Sec. 3.2 and 3.3 in Lemarié et al., 2012) with a critical value equal to $C_1$ :

$$
\lambda_s(z) = \begin{cases}
\lambda_s^{\min}, & z \leq \beta_{\min} h_{bl} \\
\displaystyle\sum_{m=0}^{3} \alpha_m \left(\frac{z}{h_{bl}}\right)^m, & z \in ]\beta_{\min} h_{bl}; \beta_{\max} h_{bl}[ \\
\lambda_s^{\max}, & z \geq \beta_{\max} h_{bl}
\end{cases}
\tag{9}
$$

where four $\alpha_m$ coefficients are necessary to guarantee the continuity of $\lambda_s(z)$ and its derivative $\partial_z \lambda_s$ at $z = \beta_{\min} h_{bl}$ and $z = \beta_{\max} h_{bl}$. We easily find

$$
\alpha_0 = \frac{(3\beta_{\max} - \beta_{\min})\beta_{\min}^2 \lambda_s^{\max} + (\beta_{\max} - 3\beta_{\min})\beta_{\max}^2 \lambda_s^{\min}}{(\beta_{\max} - \beta_{\min})^3}, \qquad \alpha_1 = -\frac{6\beta_{\max}\beta_{\min}(\lambda_s^{\max} - \lambda_s^{\min})}{(\beta_{\max} - \beta_{\min})^3}
$$
$$
\alpha_2 = 3\frac{(\beta_{\max} + \beta_{\min})(\lambda_s^{\max} - \lambda_s^{\min})}{(\beta_{\max} - \beta_{\min})^3}, \qquad \alpha_3 = -\frac{2(\lambda_s^{\max} - \lambda_s^{\min})}{(\beta_{\max} - \beta_{\min})^3}
$$

The value of $h_{bl}$ is bounded beforehand to guarantee that at least 3 grid points are such that $z \leq \beta_{\min} h_{bl}$ and $z \geq \beta_{\max} h_{bl}$. A typical profile of the $\lambda_s(z)$ is shown in Fig. 1a.

When the model is forced by the large-scale pressure gradient (or the geostrophic winds), the parameter $\lambda_m(z)$ should be theoretically zero at high and mid latitudes. However for the equatorial region, a Newtonian relaxation toward the large scale winds should be maintained. To do so, the coefficient $\lambda_m(z)$ is multiplied by a coefficient $r_{eq}$ which is a function of the Coriolis parameter $f$. The $r_{eq}$ coefficient equal to zero for large values of $|f|$ and increases to one when approaching the equator. The following form satisfies those constraints (see also Fig. 1b)

$$
r_{eq}(f) = \sin\left(\frac{\pi}{2}\left[\frac{f - f_{\max}}{f_{\max}}\right]\right)^6, \qquad f_{\max} = \frac{2\pi}{12 \times 3600} \text{ s}^{-1}.
\tag{10}
$$

## 3  Numerical discretization and implementation within NEMO

We have introduced so far the continuous formulation of the ABL1d model. In this section we describe the discretization methods used and how this model is included in the NEMO modeling framework. In particular, the discretization of the Coriolis term and of the TKE equation (6) and associated mixing lengths are described in Subsect. 3.1 and 3.2 respectively. Details about the practical implementation in NEMO are given in Subsect. 3.3 for the coupling aspects and 3.4 for the computational aspects.

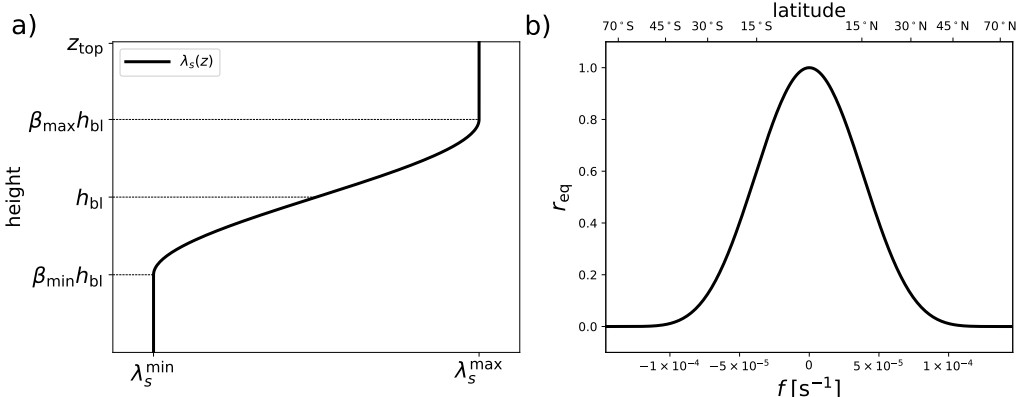

**Figure 1.** a) Typical profile of the nudging coefficient $\lambda_s(z)$ with respect to the parameters $\lambda_s^{\mathrm{max}}, \lambda_s^{\mathrm{min}}, \beta_{\mathrm{min}}, \beta_{\mathrm{max}}, h_{\mathrm{bl}}$. b) Equatorial restoring function $r_{\mathrm{eq}}$ with respect to the Coriolis frequency $f$.

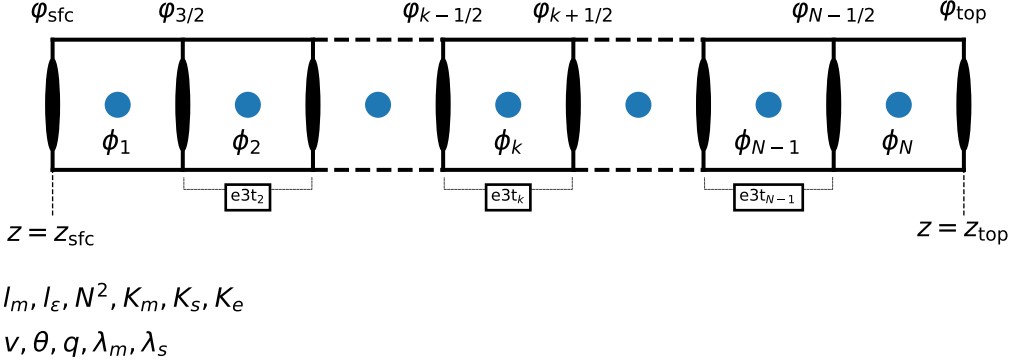

**Figure 2.** Vertical grid variable arrangements and important notations.

The ABL1d model (1) is discretized in time with an Euler backward scheme for the vertical diffusion terms, semi-implicitly for the Coriolis term and explicitly for the relaxation term which means that the model is stable as long as $\lambda_s \Delta t \leq 1$. The variables are defined on a non-staggered grid in the horizontal (a.k.a Arakawa A-grid). Because we consider a computational domain exclusively over water or sea-ice, topography is not considered and vertical levels are flat and fixed in time which, among other things, allows to interpolate the large-scale data $\phi_{\mathrm{LS}}$ on the vertical grid offline. The position of the various quantities introduced so far on the computational grid is given in Fig. 2.

## 3.1 Coriolis term treatment

Since in our implementation the horizontal velocity components are collocated, the discretization of the Coriolis term is straightforward and is energetically neutral. In the event the ABL1d is integrated with a time-step much larger than the oceanic time-step, a specific care must be given to the stability of the Coriolis term time-stepping. A semi-implicit scheme with weighting parameter $\gamma$ reads $\mathbf{u}_h^{n+1,\star} = -(f\Delta t)\mathbf{k} \times \left((1-\gamma)\mathbf{u}_h^n + \gamma\mathbf{u}_h^{n+1,\star}\right)$ where the exponent $\star$ is used here to emphasize that $\mathbf{u}_h^{n+1,\star}$ is a temporary value at time $n+1$ before vertical diffusion and Newtonian relaxation are applied. For a given grid cell with index $(i,j)$, the semi-implicit scheme can be written in a more compact way as

$$u_{i,j}^{n+1,\star} = \frac{(1-\gamma(1-\gamma)(f_{i,j}\Delta t)^2)u_{i,j}^n + (f_{i,j}\Delta t)v_{i,j}^n}{1+(f_{i,j}\Delta t)^2\gamma^2}$$

$$v_{i,j}^{n+1,\star} = \frac{(1-\gamma(1-\gamma)(f_{i,j}\Delta t)^2)v_{i,j}^n - (f_{i,j}\Delta t)u_{i,j}^n}{1+(f_{i,j}\Delta t)^2\gamma^2}.$$

The associated amplification factor modulus is $|\mathcal{A}_{\mathrm{cor}}| = \sqrt{\dfrac{1+(1-\gamma)^2(f\Delta t)^2}{1+\gamma^2(f\Delta t)^2}}$ meaning that unconditional stability is obtained as long as $\gamma \geq 1/2$ . For the numerical results obtained below in Sect. 4 and 5 we used $\gamma = 0.55$ which is deliberately slightly dissipative.

## 3.2 Discretization of TKE equation

In Sect. 2.3 we have presented the continuous formulation of the TKE-based turbulence closure of the ABL1d model. In the following we describe how the positivity of TKE can be preserved and how the mixing lengths $l_m$ and $l_\varepsilon$ are computed. We provide a substantial discussion on the latter aspect because numerical results are very sensitive to the choices made.

### 3.2.1 TKE positivity-preservation

The TKE equation is discretized using a backward Euler scheme in time with a linearization of the dissipation term $\dfrac{c_\varepsilon}{l_\varepsilon}e^{3/2}$ which is discretized as $\dfrac{c_\varepsilon}{l_\varepsilon}\sqrt{e^n}e^{n+1}$. However, such discretization is not unconditionally positivity-preserving for TKE which could give rise to unphysical solutions (e.g. Burchard, 2002b). Ignoring the diffusion term, the TKE prognostic equation (6) can be written as an ordinary differential equation (ODE) of the form

$$\partial_t e = S(\mathbf{u}_h, N^2) - D(e,t)\,e, \qquad \text{with} \qquad S(\mathbf{u}_h, N^2) = K_m\|\partial_z\mathbf{u}_h\|^2 - K_t N^2,\ D(e,t) = \frac{c_\varepsilon}{l_\varepsilon}\sqrt{e^n} \tag{11}$$

where the last term can be seen as a damping term. For ODEs like (11) it can be shown that for an initial condition $e(0) \geq 0$ and $S(\mathbf{u}_h, N^2) \geq 0$, the solution $e(t)$ keeps the same sign as $e(0)$ whatever the sign of the damping coefficient $D(e,t)$. Assuming that $S(\mathbf{u}_h, N^2)$ and $D(e,t)$ are positive, a backward Euler discretization of the damping term in (11) would lead to $e^{n+1} = \dfrac{e^n + \Delta t S(\mathbf{u}_h, N^2)}{1+\Delta t D(e,t)}$ which preserves positivity since for $e^n \geq 0$ we obtain $e^{n+1} \geq 0$. However, there is no guarantee that the forcing term $S(\mathbf{u}_h, N^2)$ is positive in particular when the shear is weak and the stratification is large. When $S(\mathbf{u}_h, N^2)$ is negative a specific treatment (known as "Patankar trick", see Deleersnijder et al., 1997; Burchard, 2002b) is required. In

the event of a negative $S(\mathbf{u}_h, N^2)$, the idea is to move the buoyancy term from $S$ to $D$ after dividing it by $e^n$, such that $S(\mathbf{u}_h, N^2) = K_m \|\partial_z \mathbf{u}_h\|^2$ is now strictly positive and $D(e,t) = \dfrac{c_\varepsilon}{l_\varepsilon}\sqrt{e^n} + K_s \dfrac{N^2}{e^n}$. Such procedure is a sufficient condition to preserve the positivity of the TKE without ad-hoc clipping of negative values. Moreover our discretization of the shear and buoyancy terms in the TKE equation is done in an energetically-consistent way following Burchard (2002a).

### 3.2.2 Mixing lengths computation

An other challenging task when implementing a TKE scheme is the discretization of the mixing lengths. As mentioned earlier, 4 different discretizations of $l_m$ (resp. $l_\varepsilon$) have been coded. All discretizations consider the boundary conditions given in (8). The values of $l_m$ and $l_\varepsilon$ are traditionally computed from two intermediate length scales $l_{up}$ and $l_{dwn}$ which respectively correspond to the maximum upward and downward displacement of a parcel of air with a given initial kinetic energy. Once $l_{up}$ and $l_{dwn}$ have been estimated by one of the method described below, the dissipative and mixing length scale $l_m$ and $l_\varepsilon$ are computed as

$$l_m = \left( \frac{1}{2} \left\{ l_{up}^{\frac{1}{a}} + l_{dwn}^{\frac{1}{a}} \right\} \right)^a \tag{12a}$$

$$l_\varepsilon = \min\left(l_{up}, l_{dwn}\right) \tag{12b}$$

where $a \approx -\frac{3}{2}$ for CBR00 and $a \approx -\frac{6}{7}$ for CCH02 (see App. A). The impact of the weighting between $l_{up}$ and $l_{dwn}$ to compute $l_m$ can be significant for idealized experiments like the ones presented in Sect. 4.2 but for more realistic cases results are weakly sensitive and equivalent to the ones obtained with the simpler weighting $l_m = \sqrt{l_{up} l_{dwn}}$.

In the following we provide the continuous form of the various ways to compute $l_{up}$ and $l_{dwn}$ implemented in the ABL1d model. The discretization aspects are detailed in App. C.

1. **Bougeault and Lacarrère (1989) length scale**. A classical approach in atmospheric models is the use of the Bougeault and Lacarrère (1989) mixing length (see also Bougeault and André, 1986) which defines $l_{up}$ and $l_{dwn}$ as

$$\int_{z}^{z+l_{up}} N^2(s)(s-z)ds = e(z), \qquad \int_{z-l_{dwn}}^{z} N^2(s)(z-s)ds = e(z) \tag{13}$$

By construction such mixing lengths are bounded by the distance to the bottom and the top of the computational domain. It is worth noting that for $N^2 = \text{cste}$, (13) gives respectively $\dfrac{l_{up}^2 N^2}{2} = e(z)$ and $\dfrac{l_{dwn}^2 N^2}{2} = e(z)$ which is equivalent to the Deardorff (1980) length scale. In the remainder we will note $l_{BL89}$ the mixing length obtained from (13).

2. **Adaptation of NEMO's length scale**. The standard NEMO algorithm (Sec. 10.1.3 in Madec, 2012) is simple and efficient compared to (13). This algorithm is based on the Deardorff (1980) length scale $l_{D80} = \sqrt{2e(z)/N^2}$. $l_{up}$ and $l_{dwn}$ are first initialized to $l_{up} = l_{dwn} = l_{D80}$. The resulting length scales are then limited not only by the distance to the surface and to the top but also by the distance to a strongly stratified portion of the air column. This limitation amounts to control the vertical gradients of $l_{up}(z)$ and $l_{dwn}(z)$ such that they are not larger that the variations of altitude. The

resulting mixing length will be simply referred to as $l_{\mathrm{D80}}$. Note that the Taylor expansion of the integral in (13) is

$$\int_{z}^{z+l_{\mathrm{up}}} N^2(s)(s-z)ds \approx \frac{N^2(z)l_{\mathrm{up}}^2}{2} + \frac{\frac{dN^2}{dz}l_{\mathrm{up}}^3}{3} + \mathcal{O}(l_{\mathrm{up}}^4),$$

which shows that the $l_{\mathrm{D80}}$ mixing length is an approximation of $l_{\mathrm{BL89}}$ which is obtained by retaining only the leading order term in the Taylor expansion.

3. **Rodier et al. (2017) length scale**. Recently, Rodier et al. (2017) proposed a modification of the Bougeault and Lacarrère (1989) mixing length. This modification turns out to improve results for stably stratified boundary layers typical of areas covered by ice. They propose to add a shear related term to (13) such that the definition of $l_{\mathrm{up}}$ and $l_{\mathrm{dwn}}$ becomes

$$\int_{z}^{z+l_{\mathrm{up}}} \left[ N^2(s)(s-z) + c_0\sqrt{e(s)}\|\partial_s\mathbf{u}_h\| \right]ds = e(z), \qquad \int_{z-l_{\mathrm{dwn}}}^{z} \left[ N^2(s)(z-s) + c_0\sqrt{e(s)}\|\partial_s\mathbf{u}_h\| \right]ds = e(z). \qquad (14)$$

where $c_0$ is a parameter whose value should be smaller than $\sqrt{C_m/c_\varepsilon}$. The value of $c_0$ will be chosen based on numerical experiments presented in Sec. 4. In the following this mixing length will be referred to as $l_{\mathrm{R17}}$.

4. **A local buoyancy and shear-based length scale**. For the sake of computational efficiency, we have derived a local version of the Rodier et al. (2017) length scale which is original to the present paper. Under the assumption that $l_{\mathrm{up}}$ (resp. $l_{\mathrm{dwn}}$) is small compared to the spatial variations of $N^2$, $e$, and $\|\partial_z\mathbf{u}_h\|$, we end up with the following second-order equation for $l_{\mathrm{up}}$: $\frac{N^2(z)}{2}l_{\mathrm{up}}^2 + c_0\sqrt{e(z)}\|\partial_z\mathbf{u}_h\|l_{\mathrm{up}} = e(z)$, whose unique positive solution is

$$l_{\mathrm{D80}}^\star(z) = \frac{2\sqrt{e(z)}}{c_0\|\partial_z\mathbf{u}_h\| + \sqrt{c_0^2\|\partial_z\mathbf{u}_h\|^2 + 2N^2(z)}}.$$

We easily find that $l_{\mathrm{D80}}^\star = l_{\mathrm{D80}}$ for $\|\partial_z\mathbf{u}_h\| = 0$, and $l_{\mathrm{D80}}^\star = \frac{\sqrt{e(z)}}{c_0\|\partial_z\mathbf{u}_h\|}$ for $N^2 = 0$ which is consistent with the shear based length scale of Wilson and Venayagamoorthy (2015). Once $l_{\mathrm{D80}}^\star$ has been computed we apply the same algorithmic approach as in the $l_{\mathrm{D80}}$ case.

The performance of those four length scales for various physical flows is discussed in Sect. 4.

## 3.3 Coupling with ocean and sea-ice

For the practical implementation of the ABL coupling strategy within a global oceanic model, a proper coupling method is required for stability and consistency purposes (e.g. Beljaars et al., 2017; Renault et al., 2019a) and the ABL1d must have the ability to handle grid cells partially covered by sea-ice. For the coupling strategy, a so-called implicit flux coupling which is unconditionally stable (App. B in Beljaars et al., 2017) and asymptotically consistent for $\Delta t \to 0$ (Renault et al., 2019a) is used. Because vertical diffusion in ABL1d is handled implicitly in time, the boundary conditions (3) should be provided at

time $n + 1$. The implicit flux coupling amounts to discretize the boundary conditions (3) as

$$K_m \partial_z \mathbf{u}_h|_{z=z_{\text{sfc}}}^{n+1} = C_D \|\mathbf{u}_h^n(z_{\text{sfc}}) - \widetilde{\mathbf{u}}_{\text{oce}}\|(\mathbf{u}_h^{n+1}(z_{\text{sfc}}) - \widetilde{\mathbf{u}}_{\text{oce}}) \tag{15}$$

$$K_s \partial_z \phi|_{z=z_{\text{sfc}}}^{n+1} = C_\phi \|\mathbf{u}_h^n(z_{\text{sfc}}) - \widetilde{\mathbf{u}}_{\text{oce}}\|(\phi^{n+1}(z_{\text{sfc}}) - \widetilde{\phi}_{\text{oce}}) \tag{16}$$

where $\widetilde{\mathbf{u}}_{\text{oce}}$ and $\widetilde{\phi}_{\text{oce}}$ are either the instantaneous values at time $n$ if NEMO and ABL1d have the same time-step or an average over the successive oceanic substeps otherwise.

A particular care has also been given to the compatibility between the ABL1d model and SI$^3$ (Sea Ice model Integrated Initiative) the sea-ice component of NEMO. SI$^3$ is a multi-category model whose state variables relevant for our study are the ice surface temperature $T_l^{\text{ice}}$ with associated fractional area $a_l$ (for the $l^{\text{th}}$ category), and the ice velocity $\mathbf{u}^{\text{ice}}$ (same for all categories). Note that the values of the exchange coefficients over sea-ice $C_D^{\text{ice}}$, $C_\theta^{\text{ice}}$, and $C_q^{\text{ice}}$ are different from their oceanic counterparts but are the same over all sea-ice categories. At this point there are several strategies for the ABL1d/SI$^3$ coupling:

1. run the ABL1d model over the whole ABL for each category $l$ and then average atmospheric variables weighted by $a_l$,

2. run a single ABL1d model with a category averaged surface flux. In the current version of NEMO $C_\theta^{\text{ice}}$ is function of the averaged temperature $T^{\text{ice}}$ which means that it is equivalent to compute a flux over each category before averaging them and to compute a single flux using the averaged surface temperature, indeed

$$\sum_l a_l \left[ C_\theta^{\text{ice}} \|\mathbf{u}_h(z_{\text{sfc}}) - \mathbf{u}^{\text{ice}}\|(\theta(z_{\text{sfc}}) - T_l^{\text{ice}}) \right] = C_\theta^{\text{ice}} \|\mathbf{u}_h(z_{\text{sfc}}) - \mathbf{u}^{\text{ice}}\| \left( \theta(z_{\text{sfc}}) - \sum_l a_l T_l^{\text{ice}} \right)$$

The second option has been preferred because it is much easier to implement and more computationally efficient. It amounts to consider an ice surface temperature averaged over all categories $T^{\text{ice}} = \sum_{l=1}^{n_{\text{cat}}} a_l T_l^{\text{ice}}$ for the computation of ice-atmosphere turbulent fluxes ($T^{\text{ice}}$ also enters in the computation of $q_{\text{ice}}$). Noting $F_{\text{oce}}$ the fraction of open water (lead), the boundary condition (15) and (16) are modified in

$$K_m \partial_z \mathbf{u}_h|_{z=z_{\text{sfc}}}^{n+1} = F_{\text{oce}} C_D \|\mathbf{u}_h^n(z_{\text{sfc}}) - \widetilde{\mathbf{u}}_{\text{oce}}\|(\mathbf{u}_h^{n+1}(z_{\text{sfc}}) - \widetilde{\mathbf{u}}_{\text{oce}}) + (1 - F_{\text{oce}}) C_D^{\text{ice}} \|\mathbf{u}_h^n(z_{\text{sfc}}) - \widetilde{\mathbf{u}}^{\text{ice}}\|(\mathbf{u}_h^{n+1}(z_{\text{sfc}}) - \widetilde{\mathbf{u}}^{\text{ice}})$$

$$K_s \partial_z \phi|_{z=z_{\text{sfc}}}^{n+1} = F_{\text{oce}} C_\phi \|\mathbf{u}_h^n(z_{\text{sfc}}) - \widetilde{\mathbf{u}}_{\text{oce}}\|(\phi^{n+1}(z_{\text{sfc}}) - \widetilde{\phi}_{\text{oce}}) + (1 - F_{\text{oce}}) C_\phi^{\text{ice}} \|\mathbf{u}_h^n(z_{\text{sfc}}) - \widetilde{\mathbf{u}}^{\text{ice}}\|(\phi^{n+1}(z_{\text{sfc}}) - \widetilde{\phi}^{\text{ice}})$$

Because the dynamics of sea-ice is computed before the thermodynamics (see Fig. 1 in Rousset et al., 2015), the ABL1d/SI$^3$ coupling follows the different steps

1. Compute surface fluxes over ice and ocean and integrate the ABL1d model for given values $F_{\text{oce}}^n$ and $a_l^n$.

2. Compute the dynamics of sea-ice

3. Update $F_{\text{oce}}^n$ and $a_l^n$ in $F_{\text{oce}}^\star$ and $a_l^\star$ because of step 2.

4. Distribute the fluxes over each ice category considering the updated values $a_l^\star$ (Sect. 3.6 in Rousset et al., 2015)

5. Compute the thermo-dynamics of sea-ice

ASL forcing strategy ABL coupling strategy

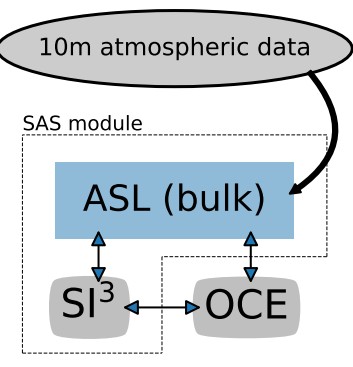 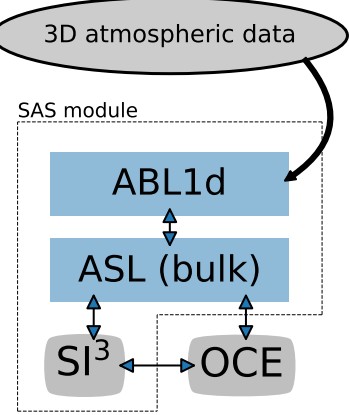

**Figure 3.** Schematic representation of the ASL forcing strategy (left) and ABL coupling strategy (right) in term of code organization and required external data. The OCE and SI$^3$ components represent respectively the oceanic and sea-ice dynamics and thermodynamics while the ASL component is in charge of providing boundary conditions related to atmospheric conditions. In the NEMO computational framework the so-called surface module (SAS component), delineated by dashed line polygons, is virtually separated from the OCE component which allows SAS to be run in standalone or detached mode (see Sec. 3.4).

## 3.4 Computational aspects

As described in Maisonnave and Masson (2015), the NEMO source code is organized to separate the ocean routines on one side and the routines responsible for the surface boundary conditions computation (including sea-ice and the coupling interfaces) on the other side. This makes a clear separation between the standard ocean model (OCE component) and the so-called surface module (SAS component). As schematically described in Fig. 3, the ABL1d model has been implemented within the SAS component which allows the following useful features

- The ABL1d model can be run in standalone mode (coupled or not with sea-ice) with prescribed oceanic surface fields.

- The ABL1d model can be run in detached mode, i.e. the OCE and SAS components run on potentially separate processors and computational grids communicating via the OASIS3 − MCT coupling library (Craig et al., 2017).

An other capability implemented within the NEMO modelling framework is the possibility to interpolate forcing fields on-the-fly. This is particularly useful for the ABL coupling strategy since three-dimensional atmospheric data must be interpolated on the ABL1d computational grid. As the current implementation of the on-the-fly interpolation only works in the horizontal, the vertical interpolation of large-scale atmospheric data on the ABL1d vertical grid is done offline. Nevertheless it means that the size of input data compared to an ASL forcing strategy is $N$ times larger with $N$ the number of vertical levels in the ABL. A possibility to improve the efficiency for the reading of input data would be to take advantage of the parallel IO capabilities

|  | Units | Neutral case | GABLS1 | SST front |
|---|---|---|---|---|
| Time-step | [s] | 60 | 10 | 10 |
| Simulation time | [h] | 28 | 9 | 40 |
| $z_{\mathrm{top}}$ | [m] | 1500 | 400 | 2000 |
| Vertical levels | - | 40 | 64 | 50 |
| Vertical resolution |  | uniform | uniform | stretched ($\Delta z \in [20, 100]$ m) |
| Coriolis parameter | [s$^{-1}$] | $10^{-4}$ | $1.39 \times 10^{-4}$ | $10^{-4}$ |
| Brunt-Väisälä frequency | [s$^{-2}$] | 0 | $3.47 \times 10^{-4}$ | $10^{-4}$ |
| Geostrophic winds | [m s$^{-1}$] | $\mathbf{u}_G = (10,0)$ | $\mathbf{u}_G = (8,0)$ | $\mathbf{u}_G = (15,0)$ |
| Roughness length | [m] | 0.1 | 0.1 | COARE3.0 bulk |
| Stability functions | - |  | $\psi_m = -4.8(z/L_{\mathrm{MO}})$ $\psi_s = -7.8(z/L_{\mathrm{MO}})$ | COARE3.0 bulk |
| $\theta_v^{\mathrm{ref}}$ | [K] | - | 283 | 288 |

**Table 2.** Description of the idealized experiments performed in Sec. 4. $L_{\mathrm{MO}}$ is the Monin-Obukhov length.

provided by the XIOS library (Xml-IO-Server, Meurdesoif et al., 2016) which is currently used in NEMO only for writing output data. This technical development is left for future work. This is a key aspect because, as discussed later in Sec. D, the main source of computational overhead associated with the ABL coupling strategy is due to the time spent waiting for input files to be read.

## 4 Atmosphere-only numerical experiments

### 4.1 Sensitivity experiments and objectives

To check the relevance of our ABL1d model for idealized atmospheric situations typical of the atmospheric boundary layer over water or sea-ice, we performed a set of single column experiments. Each of those experiments are evaluated with benchmark large-eddy simulations (LES). Moreover, we use standardized testcases from the literature to allow our results to be cross-compared with other well-established ABL schemes. In the following we consider a neutrally stratified (Sect. 4.2) and a stably stratified (Sect. 4.3) case as well as a case with a transition from stable to unstable stratification representative of an atmospheric flow over an SST front (Sect. 4.4). All ABL1d simulations presented here have been performed directly within the SAS component of the NEMO modelling framework and can be reproduced using the code available at https://zenodo.org/record/3904518 (Lemarié and Samson, 2020) which also includes the scripts to generate the figures.

An objective of the present section is to illustrate the type of sensitivity we can expect from the ABL1d model and discriminate between the various options available in the code. The experiments showed in Sect. 4.2 and 4.3 are meant to investigate the impact of (*i*) the set of constant coefficients (CBR00 vs CCH02) (*ii*) the various formulations of $l_m$ and $l_\varepsilon$ among the algorithms described in Sect. 3.2.2 (*iii*) the parameter value $c_0$ in the $l_{\mathrm{R17}}$ and $l_{\mathrm{D80}}^{\star}$ mixing length computation. Those experi-

ments will allow to discard several options. The ability of the remaining options to represent the downward mixing mechanism, discussed in Sect. 1.2, are then evaluated in Sect. 4.4. The robustness of the results to the bulk formulation and to the nudging coefficient are also checked. For each experiment we explicitly provide the initial and boundary conditions as well as all the necessary parameter values (see Tab. 2) so that the experiments can be reproduced easily by other modeling groups.

## 4.2 Neutral turbulent Ekman layer

We first propose to investigate the simulation of a neutrally stratified atmosphere analogous to a classical turbulent Ekman layer. The selected case is based on the setup described in Andren et al. (1994). The initial conditions for this experiment are not defined analytically, they are given by Tab. A.1 in Andren et al. (1994)[4]. This testcase is mainly used to check the adequacy of our surface boundary conditions with similarity theory and the proper calibration of the parameter $c_0$ in the $l_{\mathrm{D80}}^\star$ and $l_{\mathrm{R17}}$ formulations of the mixing lengths. In theory, the $l_{\mathrm{D80}}$ and $l_{\mathrm{BL89}}$ mixing lengths do not support the asymptotic limit $N^2 = 0$ but for the integrity of numerical results a minimum threshold $N_\varepsilon^2$ on the stratification is imposed in the code. In this case the procedure to compute those mixing lengths as described in App. C will provide identical results, namely $l_{\mathrm{up}} = z_{\mathrm{top}} - z$ and $l_{\mathrm{dwn}} = z - z_{\mathrm{sfc}}$ (i.e. the distance from the top and from the bottom of the computational domain). We test here the $l_{\mathrm{D80}}^\star$ and $l_{\mathrm{R17}}$ introduced in Sec. 3.2.2. The reference solution is taken from Cuxart et al. (2000) (panels a) and b) in their Fig. 16). Results are obtained using the ABL1d model with either the CBR00 (Fig. 4, left panels) or the CCH02 (Fig. 4, right panels) set of parameters. All experiments have been done with $c_0 = 0.15$ and $c_0 = 0.2$. All simulations are able to reproduce the overall behaviour of the LES case.

*Main outcomes*:

- The best agreement is obtained when using the CCH02 constants along with $l_{\mathrm{D80}}^\star$ mixing length and $c_0 = 0.2$

- The results obtained for $l_{\mathrm{D80}}$ and $l_{\mathrm{BL89}}$ are identical and close to the $l_{\mathrm{R17}}$ results with $c_0 = 0.15$ (not shown)

- All simulations with the CCH02 set of parameters show reasonable results

## 4.3 Stably stratified boundary layer (GABLS1)

Within the Global Energy and Water Exchanges (GEWEX) Atmospheric Boundary Layer Study (GABLS), idealized cases for stable surface boundary layers have been investigated (e.g. Cuxart et al., 2006). Such conditions are typical of areas covered with sea-ice. Here we consider the GABLS1 case whose technical description is available at http://turbulencia.uib.es/gabls/ gabls1d_desc.pdf. This experiment is particularly interesting as significant differences generally exist between solutions obtained from LES and single-column simulations, for example when the Bougeault and Lacarrère (1989) length scale is used (e.g. Cuxart et al., 2006; Rodier et al., 2017). A large scale geostrophic wind is imposed as well as a cooling of the surface

---

[4]However, we did not find significant differences in numerical solutions when using the following initial conditions :

$$\mathbf{u}_h(z, t = 0) = \mathbf{u}_G, \qquad e(z, t = 0) = e_{\min}$$

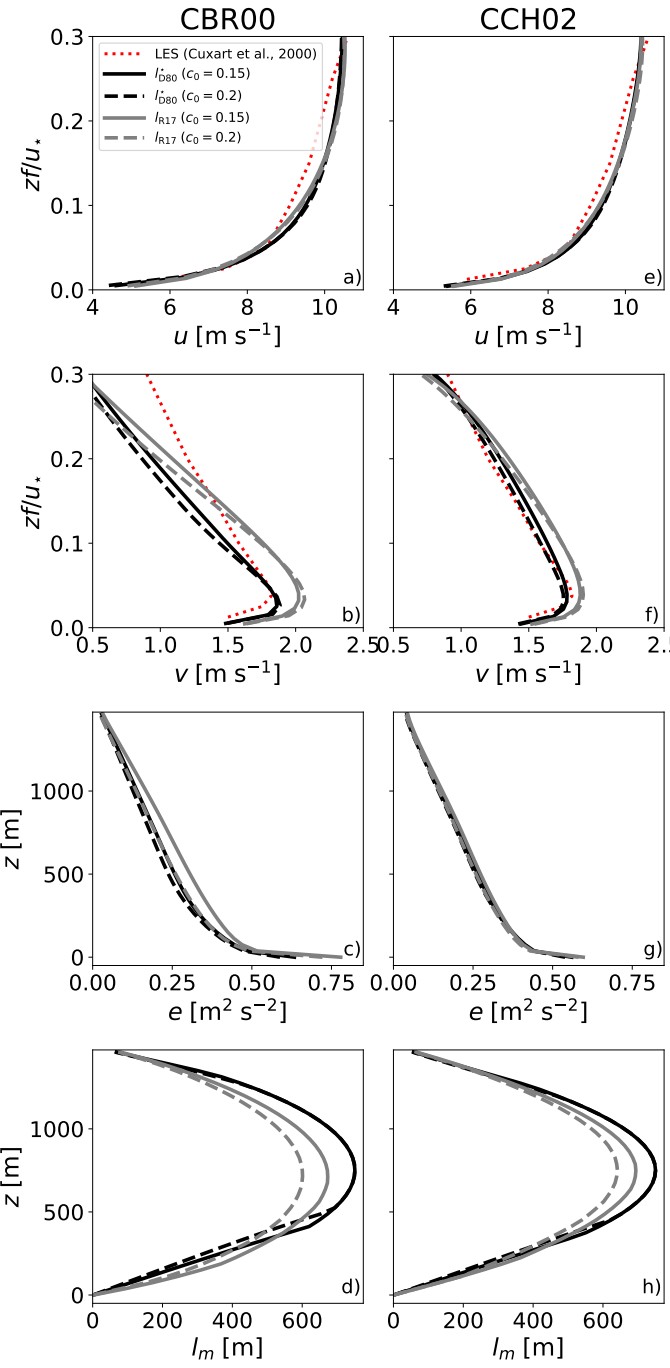

**Figure 4.** Results obtained for the neutral boundary-layer case of Andren et al. (1994) with the CBR00 model constants (left panels a to d) and CCH02 model constants (right panels d to e) for different parameter values for $c_0$ and different mixing length formulations ($l^\star_{D80}$ for black lines or $l_{R17}$ for grey lines). Results are shown for $u$ (panels a & e), $v$ (panels b & f), $e$ (panels c & g), and $l_m$ (panels d & h). In the top four panels results are compared with LES simulations from Cuxart et al. (2000) (their Fig. 16). As in Andren et al. (1994), simulations were run over a period of $10/f$ and results are averaged over the last $3/f$ period.

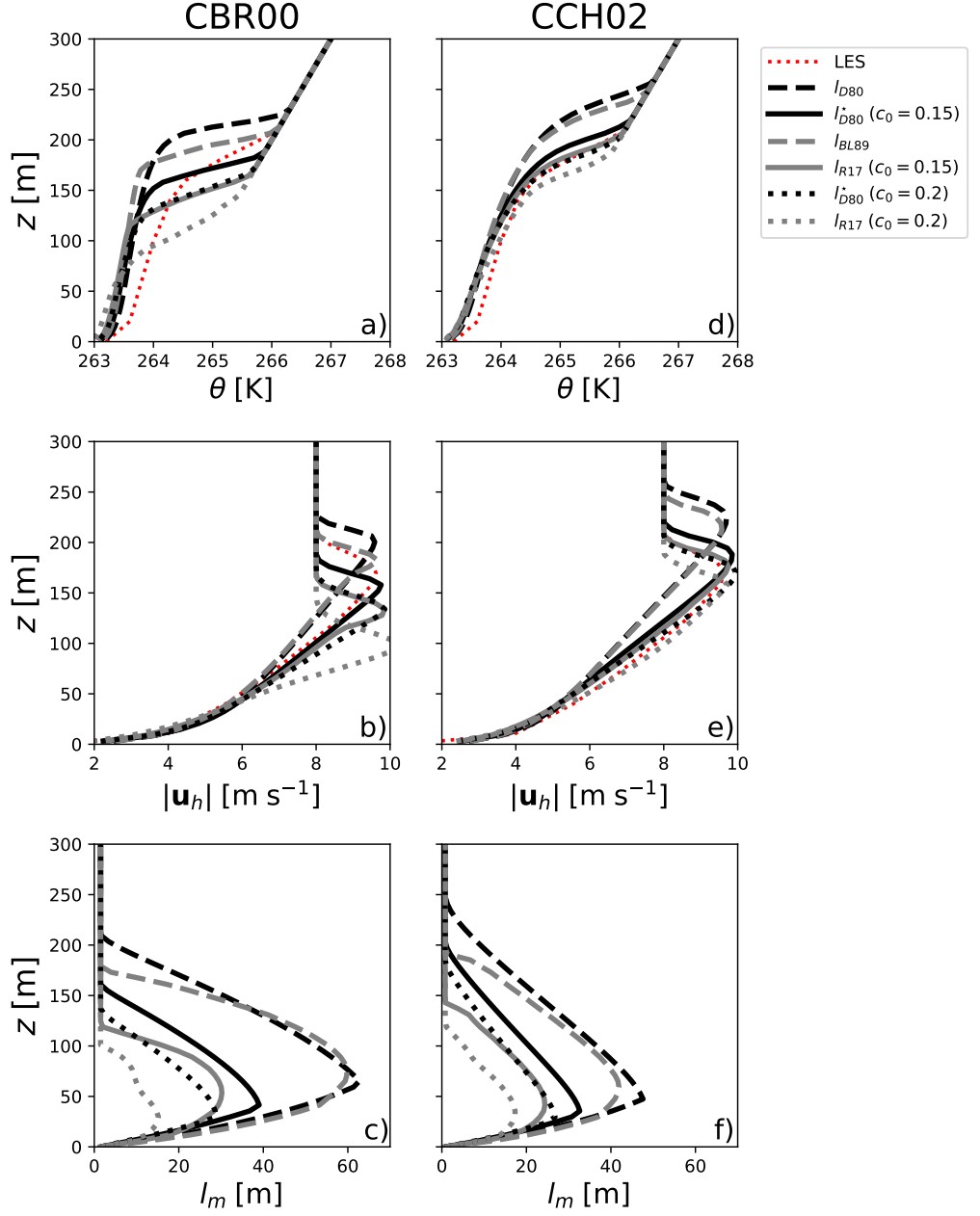

**Figure 5.** Results obtained for the stably stratified boundary-layer case of Cuxart et al. (2006) for the parameter values CBR00 (left panels a to c) and CCH02 (right panels d to f) with different mixing length formulations: $l_{D80}$ for black solid lines, $l_{D80}^{\star}$ with $c_0 = 0.15$ for dashed black lines (resp. $c_0 = 0.2$ for dotted black lines), $l_{BL89}$ for solid grey lines and $l_{R17}$ with $c_0 = 0.15$ for dashed grey lines (resp. $c_0 = 0.2$ for dotted grey lines). Results are shown for potential temperature $\theta$ (panels a & d), wind speed (panels b & e), and $l_m$ (panels c & f). Dotted red lines represent LES results from Rodier et al. (2017). Instantaneous profiles after 9 hours are shown.

temperature $\theta_s(t)$ given by $\theta_s(t) = 263.5 - 0.25(t/3600\,\mathrm{s})$. The parameter values for this test are reported in Tab. 2 and the initial conditions are $\mathbf{u}_h(z, t=0) = \mathbf{u}_G$, and

$$\theta(z, t=0) = \begin{cases} 265 & z \le 100\,\mathrm{m} \\ 265 + 0.01(z - 100) & \text{otherwise} \end{cases}, \qquad e(z, t=0) = \begin{cases} e_{\min} + 0.4(1 - z/250)^3 & z \le 250\,\mathrm{m} \\ e_{\min} & \text{otherwise} \end{cases}$$

The solutions after 9 hours of simulation are shown in Fig. 5 (left panels) for CBR00 parameter values and in Fig. 5 (right panels) for CCH02 parameter values. The reference solution is taken from Rodier et al. (2017) LES simulations. As expected, solutions based on a mixing length ignoring the contribution from the vertical shear exhibit a too thick boundary layer and a wind speed maximum located too high in altitude. Using a buoyancy and shear based mixing length mitigates the issue and provides very good agreement with reference solutions when the CCH02 model constants are used. The best results are obtained for $l_{\mathrm{D80}}^\star$ with $c_0 = 0.2$ and $l_{\mathrm{R17}}$ with $c_0 = 0.15$. Solutions obtained with the CBR00 model constants systematically predict larger turbulent kinetic energy and mixing lengths resulting in large values of $K_s$ in the first 100 meters near the surface (not shown). The mismatch in terms of TKE is partially explained by the difference in boundary condition since with CBR00 constants we have $e_{\mathrm{sfc}} = 4.628\,u_\star^2$ while with CCH02 constants we get $e_{\mathrm{sfc}} = 3.065\,u_\star^2$ from Eq. (7). Note that the proper calibration of the $c_0$ constant jointly with the $c_\varepsilon$ is the subject of several ongoing studies. Since our simulations reproduce the known sensitivity to those parameters, the ABL1d model could directly benefit from new findings on that topic.

*Main outcomes*:

- the CCH02 set of parameters provide results of better quality than the CBR00 constants. For the sake of simplicity, we will retain only the CCH02 parameters for the numerical results shown in the remainder.

- the buoyancy and vertical shear based mixing lengths $l_{\mathrm{R17}}$ and $l_{\mathrm{D80}}^\star$ are superior to the buoyancy based mixing lengths $l_{\mathrm{D80}}$ and $l_{\mathrm{BL89}}$ for stable boundary layers.

## 4.4 Winds across a midlatitude SST front

### 4.4.1 Setup and reference solutions

An idealized experiment particularly relevant for the coupling of the MABL with mesoscale oceanic eddies (and potentially submesoscale fronts) has been initially suggested by Spall (2007) and then revised by Kilpatrick et al. (2014). More recently Ayet and Redelsperger (2019) derived an analytical model based on a similar setup. The geometry of the problem is two-dimensional $x$-$z$ with an SST front along the $x$-axis

$$\theta_s(x) = 288.95 + \frac{\Delta\theta}{2}\tanh\left(\frac{x}{L_\theta}\right)$$

where $\Delta\theta = 3\,\mathrm{K}$, $L_\theta = 100\,\mathrm{km}$, and $x \in [-1800\,\mathrm{km}, 1800\,\mathrm{km}]$. As indicated in Tab. 2, a zonal geostrophic wind of $15\,\mathrm{m\,s^{-1}}$ is prescribed balanced by a vertically homogeneous meridional pressure gradient. The wind thus flows over cold water before

reaching a warm SST anomaly 3 K warmer. We consider a dry case, the model is initialized $\forall x$ with

$$\theta(z, t=0) \quad = \quad 288.95 + \left(N^2 \theta^{\mathrm{ref}}/g\right) z$$
$$q(z, t=0) \quad = \quad 0$$

where $N^2 = 10^{-4}\,\mathrm{s}^{-2}$ and $\theta^{\mathrm{ref}} = 288$ K. The velocities are systematically initialized with geostrophic winds. All simulations are run for 36 hours when the flow reaches a quasi-equilibrium state.

For this configuration the reference solution is obtained from the Mesoscale Non Hydrostatic model (MesoNH) v5.3.0 (Lafore et al., 1998; Lac et al., 2018) where microphysics and radiation packages have not been activated. The horizontal resolution is $\Delta x = 1\,\mathrm{km}$ and the model is discretized with 91 vertical levels from the surface to 20 km height. The vertical resolution near the surface is $\Delta z = 10\,\mathrm{m}$ and around $\Delta z = 100\,\mathrm{m}$ at 2000 m height. The turbulence scheme is the 1.5-order closure of Cuxart et al. (2000) in its one-dimensional form with the $l_{\mathrm{BL89}}$ mixing length and CCH02 set of parameters. Sea surface fluxes are computed using the bulk parameterisation COARE3.0. which is also available in NEMO from the Aerobulk package. As far as the ABL1d model is concerned, the top of the computational domain is $z_{\mathrm{top}} = 2000\,\mathrm{m}$ and the vertical grid is stretched with a typical resolution of 20 m near the surface and 100 m near $z = z_{\mathrm{top}}$ with a first grid point located at $z = 10\,\mathrm{m}$. In the horizontal, the resolution is $\Delta x = 6\,\mathrm{km}$.

### 4.4.2   Numerical results

For this configuration, results will be mostly evaluated in terms of 10 m winds $\mathbf{u}_{10}$ and temperature $\theta_{10}$. As an illustration of the type of result we get, we first compare the MesoNH solution and the ABL1d solution obtained with the $l_{\mathrm{D80}}$ and $l_{\mathrm{BL89}}$ mixing-lengths in Fig. 6. It is worth noting that the MesoNH solution closely compares with the solution of Kilpatrick et al. (2014) (their Fig. 2) with a shallow boundary layer height (around 400 m) before the front and a thicker one (around 800 m) after the front where momentum mixing is enhanced. Over the front, as noted by Ayet and Redelsperger (2019) with similar setup, the effect of advection is predominant for meridional winds thus explaining the differences seen with the ABL1d simulations. Indeed with ABL1d, whatever the numerical options, the atmospheric column will locally adjust to the underlying oceanic conditions since horizontal advection is neglected. This explains the absence of horizontal lag when passing over the front in the ABL1d solution compared to the MesoNH solution. However, away from the SST front the solutions are very similar in terms of boundary layer height and vertical wind structure. In anticipation of a coupling with an oceanic model, the most important quantities to look at are the 10m atmospheric variables rather than the full 3D vertical structure of the MABL. In Fig. 7, the 10m wind components and temperature when the ABL1d model reaches a quasi-equilibrium state are shown for different mixing lengths options, as well as the MesoNH results. First the results obtained with the $l_{\mathrm{R17}}$ are very different from the expected behaviour and we will focus the discussion on other options. In terms of zonal 10m wind the buoyancy based $l_{\mathrm{BL89}}$ and $l_{\mathrm{D80}}$ mixing lengths provide a good agreement with the MesoNH solution which could be expected as the MesoNH solution has been generated using the $l_{\mathrm{BL89}}$ mixing length. As soon as the mixing length is function of buoyancy and vertical shear (as is the case for $l_{\mathrm{D80}}^{\star}$) the simulated winds are weaker because the boundary layer is thinner. This leads to improved results in the stably stratified case shown earlier but in the present case more representative of realistic configuration in the

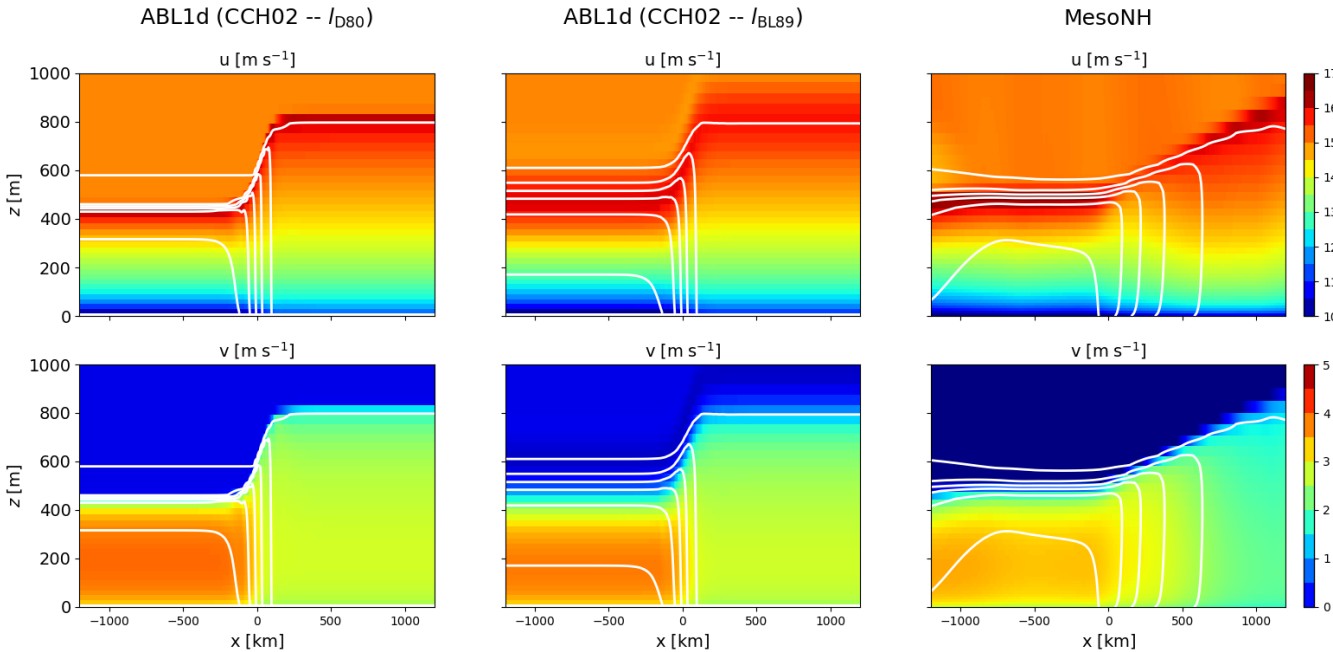

**Figure 6.** Zonal (top panels) and meridional (bottom panels) components of atmospheric winds for the reference MesoNH simulation (right panels) and for ABL1d simulations with $l_{\mathrm{D80}}$ mixing length and CCH02 model constants (left panels) and with $l_{\mathrm{BL89}}$ mixing length and CCH02 model constants (middle panels). Temperature contours are shown in white with a contour every $0.5^{\circ}$C between $15^{\circ}$C and $17.5^{\circ}$C. The SST front is centered at $x = 0$ km.

MABL it leads to a too weak mixing. However, compared to the $l_{\mathrm{R17}}$ mixing length the $l_{\mathrm{D80}}^{\star}$ still performs reasonably well but the winds on the warm side of the front are about $1\,\mathrm{m\,s^{-1}}$ weaker than the MesoNH winds for $c_0 = 0.15$ and become weaker and weaker as $c_0$ increases.

*Main outcomes*:

- in the frontal region the effect of horizontal advection is predominant and the ABL1d model can not reproduce the horizontal lag seen in the reference solution when passing over the front.

- the ABL1d model reproduces correctly the downward momentum mixing mechanism. The best results are obtained with the buoyancy based $l_{\mathrm{BL89}}$ and $l_{\mathrm{D80}}$ mixing lengths.

- the $l_{\mathrm{R17}}$ mixing length will be discarded from the comparison.

Although relevant for the present study this 2D $x$-$z$ setup is not fully representative of realistic conditions because the air column has time to adjust to the underlying oceanic state which is kept frozen in time.

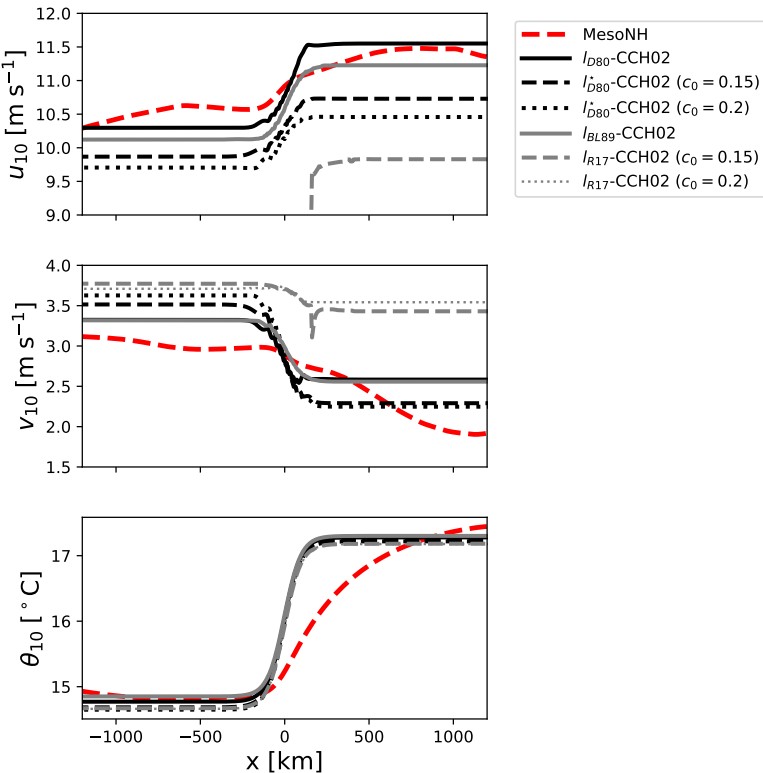

**Figure 7.** Zonal (top) and meridional (middle) components of $10\,\mathrm{m}$ winds and $10\,\mathrm{m}$ temperature (bottom) for the reference MesoNH simulation (dashed red) and for ABL1d simulations with different mixing lengths formulations for the winds across a midlatitude SST front experiment.

### 4.4.3 A single-column version

An alternative to the $x$-$z$ setup would be to formulate the testcase as a Lagrangian advection of an air column over an SST front by prescribing a temporal evolution of sea surface temperature $\theta_s(t)$ as

$$\theta_s(t) = 288.95 + 3\left[\frac{1}{2}\tanh\left(\frac{3(t - 144 \times 10^3\,\mathrm{s})}{20000}\right)\right], \qquad t \in [0, 80 \times 3600\,\mathrm{s}]$$

In this case the air column does not necessarily have time to adjust to the underlying oceanic conditions. Initial conditions are the same as the ones of the 2D $x$-$z$ case. For this testcase we do not have a reference solution but it is expected that the temporal evolution of the solution should be relatively similar to the spatial evolution in the MesoNH 2D $x$-$z$ case studied in previous subsection. This can be seen from Fig. 8 where there is a clear similarity between the time vs height sections obtained with the ABL1d simulations and the $x$ vs height sections shown for MesoNH in Fig. 6. The ABL1d solution shows a temporal lag analogous to the horizontal lag in the reference solution for the 2D $x$-$z$ case. In addition to that, we also use this testcase to investigate the sensitivity of the solutions to the bulk formulation and to the Newtonian relaxation which was absent in

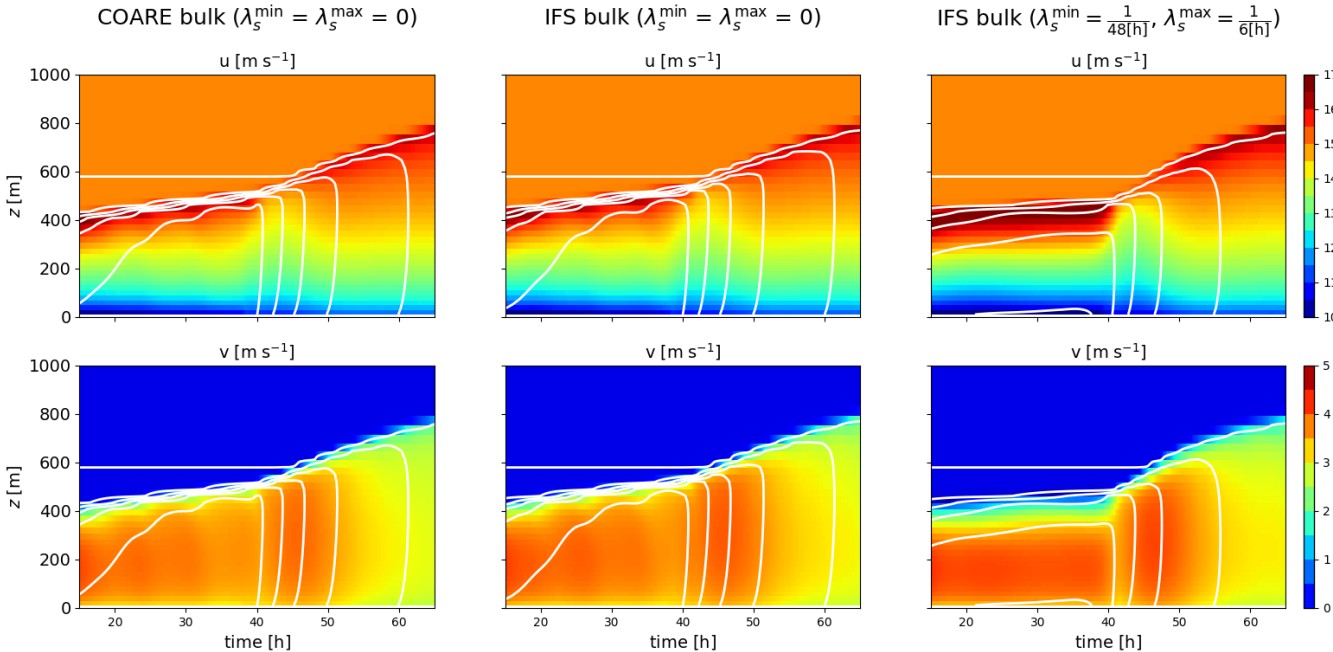

**Figure 8.** 2D time vs height sections representing the temporal evolution of the zonal (top panels) and meridional (bottom panels) components of atmospheric winds for ABL1d simulations of an air column crossing an SST front with COARE bulk formulation (left panels) and IFS bulk formulation (middle and right panels). For the case presented in the right panels, a Newtonian relaxation toward the initial temperature profile was added with $\lambda_s^{\min} = \frac{1}{48[h]}$ and $\lambda_s^{\max} = \frac{1}{6[h]}$. The simulations were performed with $l_{D80}$ mixing length and CCH02 model constants.

simulations discussed so far. We consider the COARE and IFS bulk formulation which are relatively close to each other to
check the robustness of the results to small perturbations in surface fluxes. We also consider simulations with a relaxation of
the temperature variable toward the initial condition with a fast restoring time scale $\lambda_s = \frac{1}{6[h]}$ above the PBL and a slower one
$\lambda_s = \frac{1}{48[h]}$ in the PBL. This is meant to check that the relaxation does not completely overwrites the physics of the coupling we
aim at representing with the ABL1d model. Results from those sensitivity experiments are shown in Fig. 8 and 9. In particular
in Fig. 9 the evolution of the 10 m winds across the SST front closely resembles the one shown in Fig. 7 (dashed red lines) for
MesoNH. Moreover the results in Fig. 9 are robust to a change of bulk formulation to compute the surface fluxes. Reassuringly,
adding a relaxation toward large-scale data which did not see the SST front does not deteriorate the realism of the solutions, as
can be seen from Fig. 8 (rightmost panels) and 9 (gray lines).

   ***Main outcomes***:

   – the response of the ABL1d model to evolving oceanic conditions is not local in time (it shows a temporal lag)

– the good representation of the downward momentum mixing process is not sensitive to the bulk formulation

   – adding a relaxation term toward large-scale data does not deteriorate significantly the realism of the solutions

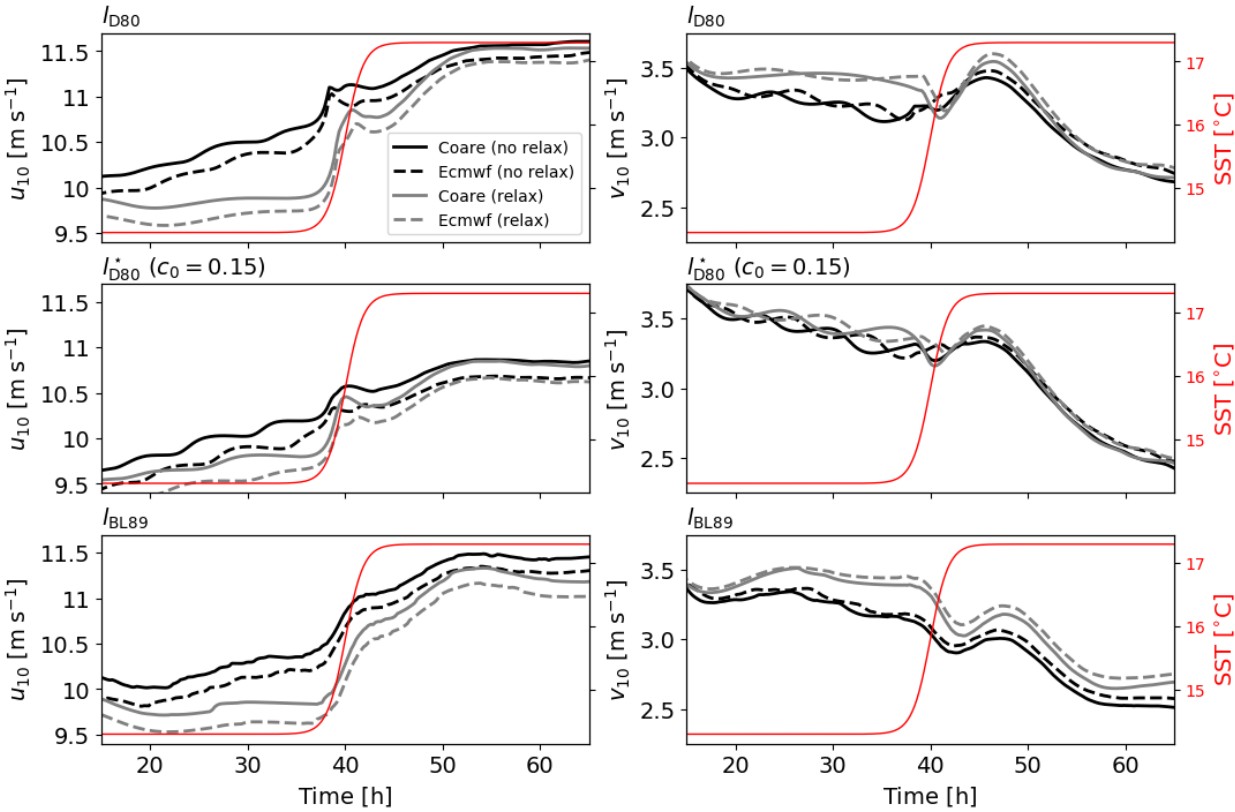

**Figure 9.** Temporal evolution of the zonal (left panels) and meridional (right panels) components of 10m atmospheric winds for ABL1d simulations of an air column crossing an SST front. The temporal evolution of SST (solid red lines) is also shown. For each panels the results from 4 different simulations are shown: with COARE bulk formulation (solid lines) or IFS bulk formulation (dashed lines), with Newtonian relaxation on temperature such that $\lambda_s^{min} = \frac{1}{48[h]}$ and $\lambda_s^{max} = \frac{1}{6[h]}$ (gray lines) or no relaxation (black lines). The top panels are obtained from simulations performed with $l_{D80}$ mixing length, middle panels with $l_{D80}^\star$ ($c_0 = 0.15$) and bottom $l_{BL89}$.

Based on the results reported in this section, the best balance between efficiency and physical relevance is obtained when using the parameter values from CCH02 and the modified Deardorff (1980) mixing length formulation $l_{D80}$ or $l_{D80}^\star$. In particular we could imagine using the $l_{D80}$ formulation over water and the $l_{D80}^\star$ formulation over sea-ice.

## 570   5   Coupled numerical experiments

Using atmosphere-only experiments, we have been focused so far on the good representation of the downward momentum mixing mechanism and of the stable boundary layers typical of areas covered with sea-ice. In the following we check that those two aspects are still adequately represented in a realistic coupled NEMO-ABL1d simulation. This simulation will also

be used to look at the wind-current interaction which was left aside so far. We performed a 5-years global simulation using the
ORCA025 configuration. Details and illustrations are given hereafter.

## 5.1 Coupled NEMO-ABL1d configuration

We use here a global ORCA025 configuration at a $0.25°$ horizontal resolution (Barnier et al., 2006) with 75 vertical $z$-levels forced by ECMWF ERA-Interim 6-hours analysis (Dee et al., 2011). This configuration is identical to the one described in Couvelard et al. (2019) (see their Sec. 4.1.1). The ABL1d-NEMO coupled simulation is carried out with the same numerical options as in a standard ASL forcing strategy. However, in the ABL coupling strategy, the two-dimensional near-surface air temperature, humidity and winds used in the usual ASL forcing are replaced by three-dimensional atmospheric variables sampled between the surface and $2000\,\mathrm{m}$ preprocessed following the different steps described in App. B. The large-scale pressure gradient computed during the preprocessing is used as a geostrophic forcing for the ABL1d model dynamics. Three-dimensional atmospheric variables are generated over the 2014-2018 period and vertically interpolated on 50 levels between $10\,\mathrm{m}$ and $2000\,\mathrm{m}$ with a vertical resolution increasing with height. Grid resolution is about $20\,\mathrm{m}$ near the air-sea interface and reaches $70\,\mathrm{m}$ at the top of the ABL1d domain. The choice of a vertical extent of $2000\,\mathrm{m}$ and 50 vertical levels in the ABL1d model is somewhat arbitrary and the robustness of numerical results to these choices will be investigated in future studies. For the simulations presented here, same horizontal grid and time-step ($\Delta t = 1200\,\mathrm{s}$) are chosen in the ABL1d and NEMO models. The options associated with the ABL coupling available through the NEMO standard namelist are reported in Tab. 3 and a detailed profiling of the code is presented in App. D in order to assess the overhead associated with the ABL coupling strategy vs ASL coupling strategy. This profiling shows that the overhead associated with the ABL1d (when using the $l_{\mathrm{D80}}$ mixing length) is in the order of $4\%$ and the one associated with the input part of the I/O operations is $5\%$. Overall there is an increase of $9\%$ in elapsed time compared to the standard ASL forcing strategy.

## 5.2 Numerical results

In this section, we evaluate the ABL coupling strategy in a realistic context for a set of relevant metrics. The objective is not to conduct a thorough physical analysis of the numerical results but to illustrate the potential of the ABL coupling strategy and its proper implementation in NEMO. To evaluate our numerical results, we use standard metrics from the literature to quantify the wind/SST (a.k.a. thermal feedback effect), wind/currents (a.k.a. current feedback effect) and MABL/sea-ice couplings (e.g. Bryan et al., 2010; Renault et al., 2019b).

### 5.2.1 Thermal feedback effect

To quantify the surface wind response to SST, we show in Fig. 10a a global map of the temporal correlation between the high-pass filtered $10\,\mathrm{m}$ wind speed from the first vertical level in the ABL1d model and the SST. Same correlation is shown in Bryan et al. (2010) from satellite observations (their Fig. 1d) and from coupled numerical experiments between a $0.1°$ ocean and a $0.25°$ atmospheric models (their Fig. 1c). Consistent with observations and fully-coupled models, the correlation

| Namelist parameter | Type | Description |
|---|---|---|
| ln_hpgls_frc | boolean | true if $\mathbf{R}_{\mathrm{LS}} = \left(\frac{1}{\rho_a}\nabla_h p\right)_{\mathrm{LS}}$ in (1) |
| ln_geos_winds | boolean | true if $\mathbf{R}_{\mathrm{LS}} = f\mathbf{k} \times \mathbf{u}_G$ in (1) |
| nn_dyn_restore | integer | (=0) no wind relaxation (=1) wind relaxation scaled by $r_{\mathrm{eq}}(f)$ as in (10) (=2) wind relaxation everywhere |
| rn_ldyn_min | real | inverse of $\lambda_m^{\max}$ in hours (see Sec. 2.4) |
| rn_ldyn_max | real | inverse of $\lambda_m^{\min}$ in hours (see Sec. 2.4) |
| rn_ltra_min | real | inverse of $\lambda_s^{\max}$ in hours (see Sec. 2.4) |
| rn_ltra_max | real | inverse of $\lambda_s^{\min}$ in hours (see Sec. 2.4) |
| nn_amxl | integer | (=0) $l_{\mathrm{D80}}$ mixing length (=1) $l_{\mathrm{D80}}^{\star}$ mixing length (=2) $l_{\mathrm{BL89}}$ mixing length (=3) $l_{\mathrm{R17}}$ mixing length |
| rn_Cm | real | $C_m$ parameter in TKE scheme |
| rn_Ct | real | $C_s$ parameter in TKE scheme |
| rn_Ce | real | $C_e$ parameter in TKE scheme |
| rn_Ceps | real | $c_\varepsilon$ parameter in TKE scheme |
| rn_Rod | real | $c_0$ parameter in $l_{\mathrm{D80}}^{\star}$ and $l_{\mathrm{R17}}$ mixing lengths |
| rn_Ric | real | $C_1$ parameter in the definition of $\phi_z$ |
| ln_smth_pblh | boolean | horizontal smoothing of PBL height |

**Table 3.** Namelist parameters in the NEMO(v4.0) to set in the namelist section namsbc_abl before running a simulation coupled with ABL1d.

obtained from the coupled NEMO-ABL1d simulation shows large positive correlations over regions like the Southern Ocean, Kuroshio and Gulf Stream extensions as well as in the Gulf of Guinea. Correlations are however weaker than observations in the northern and equatorial Pacific between $90°$ W and $180°$ W. As the thermal feedback strength is related to the ocean model resolution (Bryan et al., 2010), we can expect a better agreement with observations using a higher resolution configuration such as ORCA12 ($1/12°$ resolution). This coupling sensitivity to the oceanic resolution will be addressed in a future study.

### 5.2.2 Current feedback effect

Other processes of interest are those related to the coupling between oceanic surface currents, wind-stress and wind. Such coupling is responsible for a dampening of the eddy mesoscale activity in the ocean. In Renault et al. (2019b), two coupling coefficients called $s_w$ and $s_\tau$ are defined to quantify this effect. $s_\tau$ is a measure of the sink of energy from the eddies and fronts to the ABL and $s_w$ quantifies the partial re-energization of the ocean by the wind response to the wind/current coupling. This re-energization is absent in the ASL forcing strategy which results in an excessive dampening of the oceanic eddy mesoscale activity. In practice, $s_\tau$ (resp. $s_w$) corresponds to the slope of the linear relationship between high-pass filtered surface current vorticity and surface wind-stress (resp. wind) curl. Global maps of $s_\tau$ and $s_w$ computed from our coupled NEMO-ABL1d global simulation are shown in Fig. 10. Large negative values of $s_\tau$ indicate an efficient dampening of the eddy mesoscale

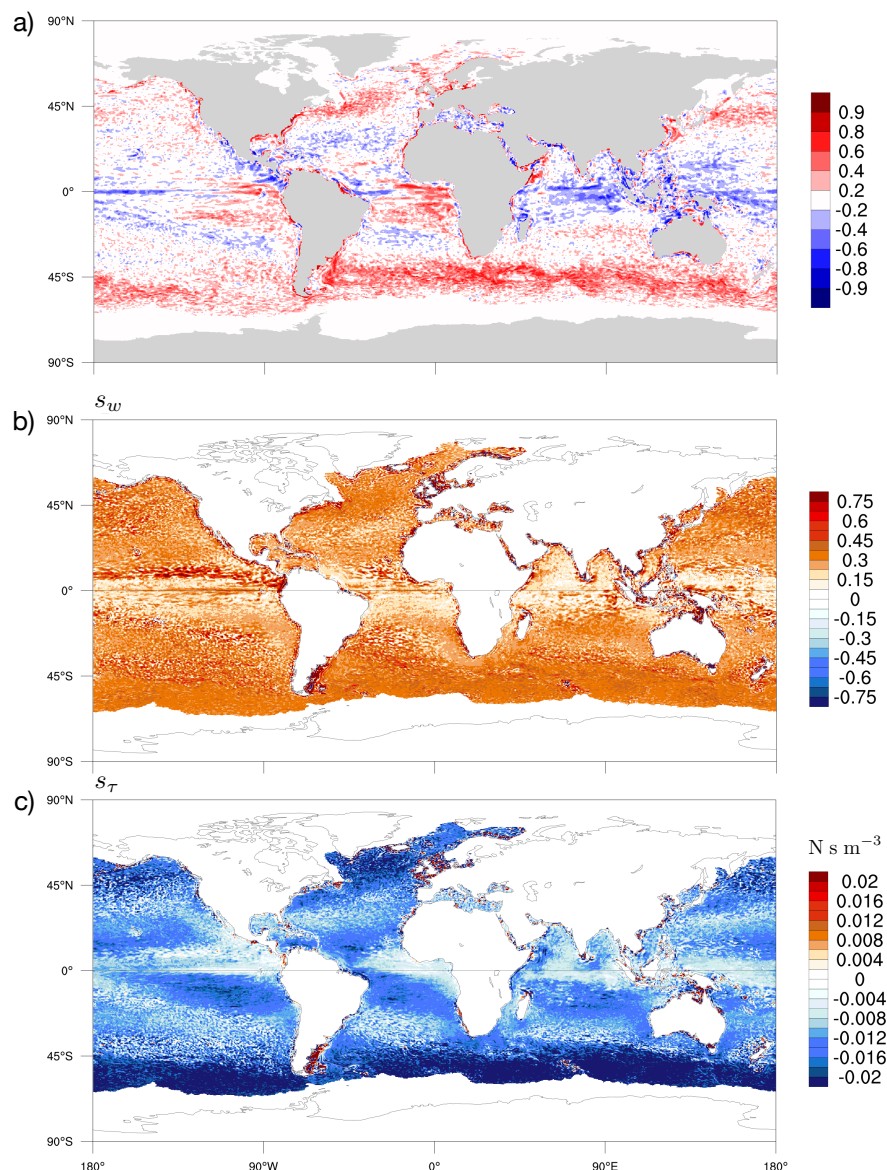

**Figure 10.** (a): Global map of temporal correlation of high-pass filtered wind speed at the first vertical level of the ABL1d model with SST from NEMO. Both NEMO and ABL1d are configured at $0.25°$ resolution. (b) & (c): Global maps of the coupling coefficient between the surface current vorticity and the wind curl ($s_w$, (b)) and between the surface current vorticity and the wind-stress curl ($s_\tau$, (c)) estimated from a $0.25°$ resolution coupled NEMO-ABL1d global simulation. The fields are first temporally averaged using a 29-day running mean and spatially high-pass filtered.

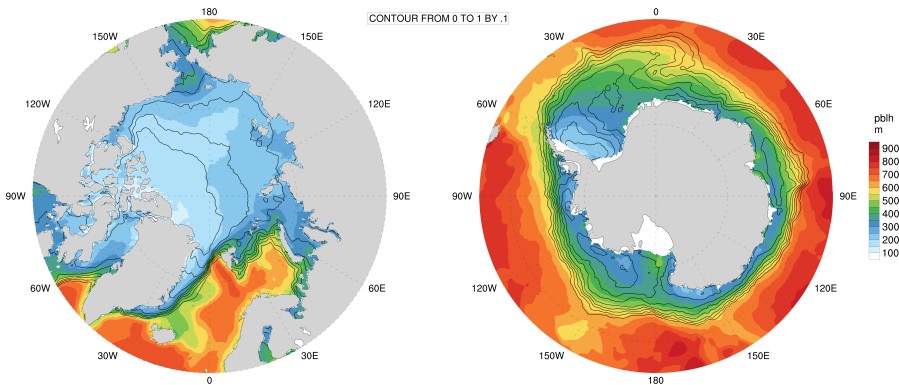

**Figure 11.** Yearly average of sea-ice cover (contours) and atmospheric boundary layer height (shaded) over the antarctic (right) and arctic (left) regions.

activity by the current feedback (i.e. a large sink of energy from the ocean to the atmosphere) and the large positive values of $s_w$ indicate an efficient wind response and re-energization of the mesoscale currents. Our numerical experiment provides results very consistent with the results obtained from coupled simulations between NEMO and the Weather and Research Forecasting model (WRF) shown in Renault et al. (2019b) (their Fig. 1b for $s_\tau$ and 2c for $s_w$). As mentioned earlier, with an ASL forcing strategy we would systematically have $s_w = 0$ and stronger $s_\tau$ values.

### 5.2.3 MABL/sea-ice coupling

The last illustration of our implementation presented in this section is the coupling of ABL1d with sea-ice. As described in Sect. 4.3, sea-ice generally induces a shallow stably stratified boundary layer due to the near-surface air cooling. This increased vertical stability tends to reduce atmospheric turbulence, producing shallower PBL heights over sea-ice. This relationship between sea-ice concentration and PBL height is clearly visible from Fig. 11 on both Arctic and Antarctic domains where the ABL height follows a progressive decrease from about 800 m to 200 m in the transition zone between the open ocean and fully ice-covered regions. This coupling between the PBL and sea-ice have important effects on near-surface wind, temperature and humidity, and consequently on sea-ice concentration evolution which will need to be specifically assessed in future ABL-based studies.

## 6 Conclusions

### 6.1 Summary

A simplified atmospheric boundary layer (ABL) model has been developed and integrated to an oceanic model. This development is made with the objective to improve the representation of air-sea interactions in eddying oceanic models compared to the standard forcing strategy where the 10 meter height atmospheric quantities are prescribed. For this preliminary study, the sim-

plified ABL model takes the form of a single column model including a turbulence scheme coupled to each oceanic grid point. A crucial hypothesis is that the dominant process at the characteristic scale of the oceanic mesoscale is the so-called downward mixing process which stems from a modulation of atmospheric turbulence by sea surface temperature (SST) anomalies. Our approach can be seen as an extended bulk approach: instead of prescribing atmospheric quantities at 10 meters to compute air-sea fluxes via an atmospheric surface layer (ASL) parameterization, atmospheric quantities in the first few hundred meters are used to constrain an ABL model which provides 10 meter atmospheric values to the ASL parameterization. An important point is that our modeling strategy keeps the computational efficiency and flexibility inherent to ocean only modeling. Indeed, the overhead generally observed in term of computational cost compared to the usual ASL forcing strategy is roughly $10\%$, half of this overhead is due to I/O operations since the ABL model is constrained by 3D atmospheric data.

In this paper the key components of such an approach have been described. This includes the large-scale forcing strategy, the coupling with the ocean and sea-ice and last but not least the ABL turbulence closure scheme based on a prognostic equation for the turbulence kinetic energy. The resulting simplified model, called ABL1d, has been tested for several boundary-layer regimes relevant to either ocean/atmosphere or sea-ice/atmosphere coupling. Results have systematically been evaluated against Large Eddy Simulations (LES). Furthermore we have investigated the behaviour of the model to several parameters including the formulation of the mixing length and the turbulence model constants. First results from a global ABL1d-NEMO configuration show an excellent behaviour in term of wind-SST two-way coupling. A first analysis of the impact of the coupling with ABL1d from a physical viewpoint is presented in Brivoal et al. (2020).

## 6.2 Future work

Now that an adequate computational framework and an efficient turbulent scheme that can be operated for a reasonable computational overhead have been developed, the next step is to investigate the relevance of the single-column representation of the ABL selected for the present paper. Indeed, several studies have already shown that momentum vertical turbulent mixing, pressure gradient, Coriolis, and nonlinear advection are all important to the momentum balance in the marine atmospheric boundary layer at the vicinity of oceanic fronts (see for example Spall, 2007; Small et al., 2008; O'Neill et al., 2010). It is well known that the relative importance of those terms depends on the wind regime: for strong winds the vertical mixing is the dominant mechanism while for weak winds the pressure adjustment mechanism dominates. The current single-column approximation is based on the assumption that the balance is dominated by vertical turbulent mixing and the effect of other terms is roughly represented by the geostrophic guide and/or the nudging term. The testcase presented in Sect. 4.4 clearly illustrates the limitations of a single-column approach ignoring advective effects. However, before moving to more advanced formulations, our rationale was that the two main bottlenecks in term of computational cost inherent to the ABL coupling strategy are the reading of 3D atmospheric data and the choice of ABL turbulent scheme. As a first step, we focused on those two aspects to assess whether or not our approach can be a viable option. Even if the justification of our model is not beyond reproach, it already brings an improvement compared to the ASL forcing strategy.

Several ways to improve the methodology presented here are currently under investigation. At a practical level, ways to lower the computational overhead due to I/O operations will be investigated using the parallel I/O capabilities provided by the

XIOS library which is currently used in NEMO only for outputs. At a more fundamental level, the continuous formulation of the ABL1d model will be completed to improve the representation of the momentum balance by integrating the effect of horizontal advection and fine-scale pressure gradient. Increasing the complexity of the model should allow to lower the impact of the nudging term on the ABL solutions.

In the event our approach turns out to be physically sound for a reasonable complexity it could be useful not only for offline oceanic simulations but also in coupled simulations to downscale the information from a low-resolution atmospheric component to a high-resolution oceanic component. A standalone ABL model of intermediate complexity could also play a role in coupled data assimilation where the current practice is generally to assimilate data separately in the ocean and the atmosphere ignoring the air-sea interactions which results in inconsistencies at the air-sea interface in the initial conditions causing initial shocks in the coupled forecasts (e.g. Mulholland et al., 2015). We wish to conclude this study by clarifying that the framework we have developed within NEMO is general enough to allow alternative approaches (e.g. via model-driven empirical models) to be seamlessly tested and confronted with the ABL coupling strategy.

*Code availability.* TEXT

*Data availability.* TEXT

*Code and data availability.* The changes to the NEMO code have been made on the standard NEMO code (release 4.0). The code can be downloaded from the NEMO website (http://www.nemo-ocean.eu/, last access: 23 June 2020). The NEMO code modified to include the ABL1d model is available in the zenodo archive (https://doi.org/10.5281/zenodo.3904518, Lemarié and Samson (2020)). The namelists and data used to produce the figures are also available in the zenodo archive.

*Sample availability.* TEXT

*Video supplement.* TEXT

## Appendix A: Surface boundary conditions for TKE and mixing lengths

In this appendix, following the methodology of Redelsperger et al. (2001) re-expressed with our notations, we quickly recall how the surface boundary conditions for the turbulent kinetic energy (TKE) and for the mixing lengths $l_m$ and $l_\varepsilon$ are determined via a matching between the subgrid turbulence scheme and the surface-layer theory. The simplest form of atmospheric surface-

layer (ASL) theory, namely for neutral stratification, is considered. Under the quasi equilibrium hypothesis the evolution equation (6) for the turbulent kinetic energy (TKE) reduces to the equilibrium between turbulence production and dissipation $K_m\|\partial_z\mathbf{u}_h\|^2 = \frac{c_\varepsilon}{l_\varepsilon}e^{3/2}$ which, combined with $K_m = C_m l_m\sqrt{e}$, leads to

$$e = \frac{C_m}{c_\varepsilon}(l_m l_\varepsilon)\|\partial_z\mathbf{u}_h\|^2 \tag{A1}$$

The similarity theory for the ASL in the neutral case is such that $\|\partial_z\mathbf{u}_h\| = \frac{u_\star}{\kappa(z+z_0)}$ with $\kappa$ the Von Karman constant and $z_0$ the roughness length which can be combined with (A1) to get that in the surface-layer:

$$e = \frac{C_m}{c_\varepsilon}\left(\frac{l_m l_\varepsilon}{\kappa^2(z+z_0)^2}\right)u_\star^2 \tag{A2}$$

with $u_\star^2 = \|\langle\mathbf{u}_h' w'\rangle\|$ the friction velocity. Moreover, enforcing the consistency between the eddy-diffusivity for momentum given by the ASL theory ($K_m = \kappa u_\star(z+z_0)$) and the one given by the TKE closure ($K_m = C_m l_m\sqrt{e}$) leads to

705 $\quad l_m = \frac{\kappa(z+z_0)}{C_m}\left(\frac{u_\star}{\sqrt{e}}\right) \qquad \rightarrow \qquad l_m^3 l_\varepsilon = c_\varepsilon C_m^{-3}\left(\kappa(z+z_0)\right)^4 \tag{A3}$

We thus have two relations (A2) and (A3) for three unknowns ($e$, $l_m$ and $l_\varepsilon$). At this point our derivation will differ from Redelsperger et al. (2001) as we will assume that $l_m = l_\varepsilon = L_{\text{sfc}}$ in the ASL. Under this assumption, combining (A2) and (A3) we easily obtain

$$e = \frac{u_\star^2}{\sqrt{C_m c_\varepsilon}}, \qquad L_{\text{sfc}}(z) = \kappa\frac{(C_m c_\varepsilon)^{1/4}}{C_m}(z+z_0).$$

for $z \le \delta_{\text{asl}}$ where $\delta_{\text{asl}}$ corresponds to the extent of the ASL.

The expression of $L_{\text{sfc}}(z)$ for $z \le \delta_{\text{asl}}$ is also used as a constraint to define the weighted average needed to determine $l_m$ from $l_{\text{up}}$ and $l_{\text{dwn}}$: $l_m = \left(\frac{1}{2}\left(l_{\text{up}}^{1/a} + l_{\text{dwn}}^{1/a}\right)\right)^a$ (equivalent to equation (12a)). In the ASL we further assume that $l_{\text{dwn}} \approx 0$ and $l_{\text{up}} \approx \delta_{\text{asl}}$, for $l_m(z = \delta_{\text{asl}})$ to be consistent with $L_{\text{sfc}}(z = \delta_{\text{asl}})$ we should have

$$\left(\frac{1}{2}\left(\delta_{\text{asl}}^{1/a}\right)\right)^a = \kappa\frac{(C_m c_\varepsilon)^{1/4}}{C_m}(\delta_{\text{asl}} + z_0) \tag{A4}$$

Considering that $\delta_{\text{asl}} \gg z_0$, (A4) is satisfied for

$$a = -\left(\frac{\log(c_\varepsilon) - 3\log(C_m) + 4\log(\kappa)}{\log(16)}\right)$$

Considering the CBR00 model constants we obtain $a = -1.4796 \approx -3/2$ and $a = -0.860834 \approx -6/7$ for the CCH02 constants (see Tab. 1).

## Appendix B: Preprocessing of atmospheric data from IFS

### B1  Altitude of IFS vertical levels

The ABL1d model is discretized on fixed in time and space geopotential levels while the IFS model uses a pressure-based sigma coordinate. A first step is to recover the altitude associated with each sigma level. The pressure $p_{k+\frac{1}{2}}$ defined at cell

interfaces between two successive vertical layers is given by

$$p_{k+\frac{1}{2}} = A_{k+\frac{1}{2}} + B_{k+\frac{1}{2}} p_s, \qquad k \in [\![1, N_{\text{ifs}}]\!]$$

where $A_{k+\frac{1}{2}}$ (Pa) and $B_{k+\frac{1}{2}}$ (dimensionless) are constants given by a smooth analytical function defining the vertical grid stretching. Typical values of the altitude of grid points in the vertical for a standard $60$ levels grid (L60) and a surface pressure of $1013\,\text{hPa}$ are given in Tab. B1.

Once the values of $p_{k+\frac{1}{2}}$ and $p_s$ are known, the altitude of cell interfaces can be computed by integrating the hydrostatic equilibrium

$$\partial_z \phi = -\frac{R_d T_v}{p} \partial_z p \tag{B1}$$

vertically. In (B1), $\phi$ is the geopotential, $T_v$ the virtual temperature and $R_d$ the specific gas constant for dry air. At a discrete level we get

$$\int_{z_{k-\frac{1}{2}}}^{z_{k+\frac{1}{2}}} \partial_s \phi \, ds = -R_d T_v(z_k) \int_{z_{k-\frac{1}{2}}}^{z_{k+\frac{1}{2}}} \left( \frac{\partial_s p}{p} \right) ds$$

which gives

$$\text{e3t}_k^{\text{ifs}} = -\frac{R_d T_v(z_k)}{g} \ln \left( \frac{p_{k+\frac{1}{2}}}{p_{k-\frac{1}{2}}} \right)$$

Once the the layer thicknesses $\text{e3t}_k^{\text{ifs}}$ are known, horizontal wind components, potential temperature and specific humidity can be interpolated on the ABL1d vertical levels. Under the constraint that $\int_{z_{\text{sfc}}}^{z_{\text{top}}} \psi^{\text{ifs}}(z) \, \mathrm{d}z = \int_{z_{\text{sfc}}}^{z_{\text{top}}} \psi(z) \, \mathrm{d}z$ for any IFS quantity $\psi^{\text{ifs}}$ to be interpolated. Wind components are interpolated using a fourth-order compact scheme while tracers are interpolated using a WENO-like PPM scheme (A. Shchepetkin, personal communication) which is monotonic.

## B2   Filtering in the presence of boundaries

Because of the IFS numerical formulation and of the post-processing of output data, the solutions sometimes contain high frequency oscillations at the vicinity of the land-sea interface. This problem is further compounded when the nearshore topography is steep. The atmospheric fields over water thus need to be smoothed horizontally to specifically remove the $2\Delta x$ noise. We use a standard two-dimensional Shapiro filter which, in the absence of lateral boundaries, can be formulated as

$$\psi_{i,j}^{\star} = \psi_{i,j} + \frac{1}{4} \left( \delta_{i+1/2,j}^{(x)} - \delta_{i-1/2,j}^{(x)} \right)$$

$$\psi_{i,j}^{\text{f}} = \psi_{i,j}^{\star} + \frac{1}{4} \left( \delta_{i,j+1/2}^{(y,\star)} - \delta_{i,j-1/2}^{(y,\star)} \right)$$

where $\delta_{i+1/2,j}^{(x)} = \psi_{i+1,j} - \psi_{i,j}$ and $\delta_{i,j+1/2}^{(y,\star)} = \psi_{i,j+1}^{\star} - \psi_{i,j}^{\star}$. The amplification factor associated to this filter is

$$\mathcal{A}_{\text{shap}}(\theta_x, \theta_y) = \frac{1}{4} (1 + \cos\theta_x)(1 + \cos\theta_y), \qquad \theta_x = k_x \Delta x, \quad \theta_y = k_y \Delta y$$

| Index | $A_k$ [Pa] | $B_k$ | Altitude $z_k$ [m] | Layer thickness $\text{e3t}_k^{\text{cep}}$ [m] |
|---|---|---|---|---|
| 1 | 0.000000 | 1.000000 | 10.00 | 20.00 |
| 2 | 0.000000 | 0.997630 | 34.97 | 29.94 |
| 3 | 7.367743 | 0.994019 | 71.89 | 43.92 |
| 4 | 65.889244 | 0.988270 | 124.48 | 61.30 |
| 5 | 210.393890 | 0.979663 | 195.85 | 81.49 |
| 6 | 467.333588 | 0.967645 | 288.55 | 104.01 |
| 7 | 855.361755 | 0.951822 | 404.72 | 128.43 |
| 8 | 1385.912598 | 0.931940 | 546.06 | 154.40 |
| 9 | 2063.779785 | 0.907884 | 713.97 | 181.61 |
| 10 | 2887.696533 | 0.879657 | 909.57 | 209.81 |
| 11 | 3850.913330 | 0.847375 | 1133.73 | 238.78 |
| 12 | 4941.778320 | 0.811253 | 1387.12 | 268.33 |
| 13 | 6144.314941 | 0.771597 | 1670.26 | 298.31 |
| 14 | 7438.803223 | 0.728786 | 1983.49 | 328.58 |

**Table B1.** Altitude $z_k$ and layer thickness $\text{e3t}_k$ of the IFS L60 vertical grid in the first 2000 meters with respect to the parameter values $A_k$ and $B_k$ of a surface pressure $p_s = 1013$ hPa.

which guarantees that one iteration of the filter is sufficient to remove the grid-scale noise since $\mathcal{A}_{\text{shap}}(\pi, \pi) = \mathcal{A}_{\text{shap}}(\pi, \theta_y) = \mathcal{A}_{\text{shap}}(\theta_x, \pi) = 0$ and that $\mathcal{A}_{\text{shap}} \leq 1$ (i.e. no waves are amplified). In the presence of solid boundaries we would like to retain those properties as much as possible. A straightforward approach would be to impose a no-gradient condition at the coast, i.e. $\delta_{i+1/2,j}^{(x)} = 0$ as soon as $\text{tmask}_{i+1,j} \times \text{tmask}_{i,j} = 0$ (resp. $\delta_{i,j+1/2}^{(y,\star)} = 0$ as soon as $\text{tmask}_{i,j+1} \times \text{tmask}_{i,j} = 0$ ) with tmask the indicator function equal to 1 over water and 0 over land. Let us also consider the following alternative boundary conditions

$$\begin{cases} \delta_{i+1/2,j}^{(x)} & = & -\delta_{i-1/2,j}^{(x)}, & \text{if } \text{tmask}_{i+1,j} = 0 \\ \delta_{i-1/2,j}^{(x)} & = & -\delta_{i+1/2,j}^{(x)}, & \text{if } \text{tmask}_{i-1,j} = 0 \end{cases} \tag{B2}$$

and similar in the $y$-direction. We do not elaborate on this choice but it can be shown theoretically that boundary conditions (B2) provide a better control of grid-scale noise near the coast. To illustrate this point, in Fig. B1 the surface pressure gradients are shown for different boundary conditions. In particular it can be seen near the coast that the no gradient boundary condition (panels b and e) leaves some artifical patterns in gradients especially in the Peru-Chile current system while boundary condition (B2) efficiently mitigate this issue. Note that it is particularly essential to make sure that the surface pressure field is sufficiently smooth because gradients of this field are used to compute geostrophic winds which are important for the large-scale forcing of the ABL1d model.

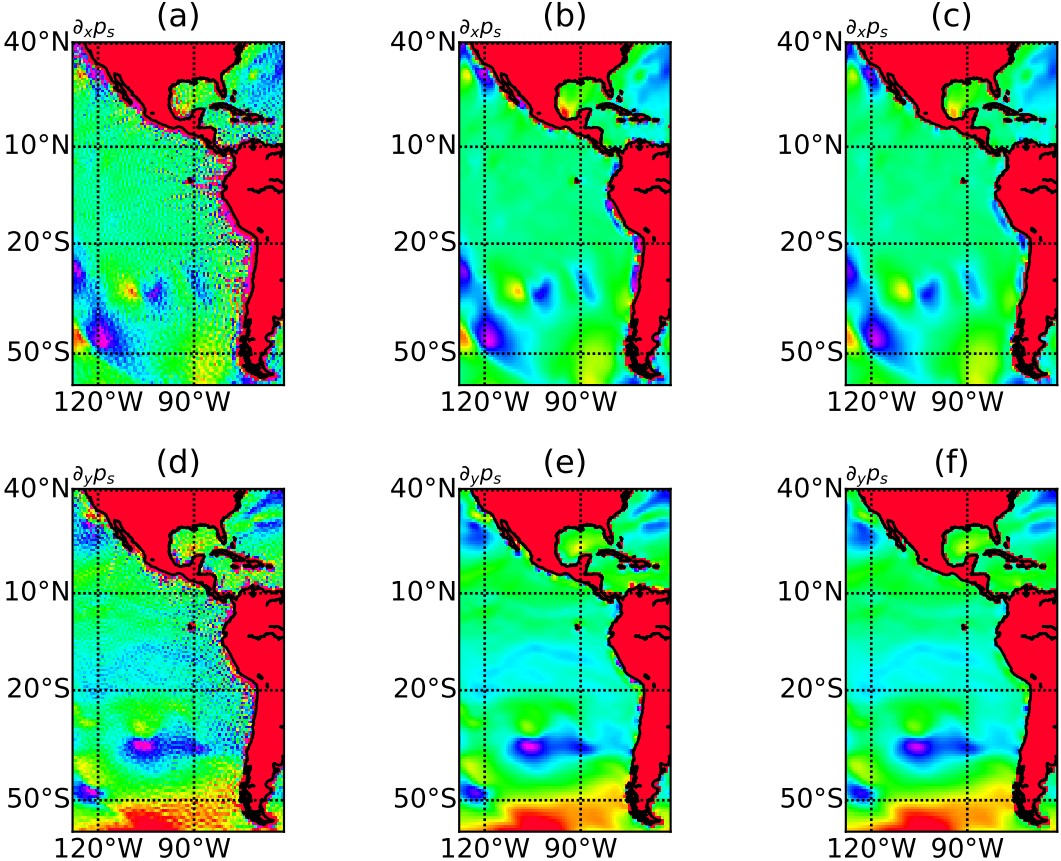

**Figure B1.** Atmospheric surface pressure horizontal gradients in $x$ (top panels) and $y$ (bottom panels) directions obtained from the original IFS data (a and d), after a Shapiro filtering with no gradient boundary conditions (b and e), and after a Shapiro filtering with boundary conditions (B2) (c and f). The area in red is covered by land.

## B3 Large-scale pressure gradient computation

The last aspect of the pre-processing of atmospheric data we would like to discuss is the computation of the large-scale pressure gradient (or equivalently of the geostrophic wind components) The objective is to estimate the following terms

$$R_{\mathrm{LS}}^{u} = \frac{1}{\rho_a}(\partial_x p)_z, \qquad R_{\mathrm{LS}}^{v} = \frac{1}{\rho_a}(\partial_y p)_z,$$

where $(\cdot)_z$ denotes a gradient along constant geopotential height. Using the hydrostatic balance we have $\frac{1}{\rho_a} = -g(\partial_z p)^{-1}$ which leads to

$$R_{\mathrm{LS}}^{u} = -g(\partial_z p)^{-1}(\partial_x p)_z, \qquad R_{\mathrm{LS}}^{v} = -g(\partial_z p)^{-1}(\partial_y p)_z \tag{B3}$$

Assuming a generalized vertical coordinate $s = s(x,y)$ the computation of gradients along constant height is not straightforward since $(\partial_x p)_z = (\partial_x p)_s - (\partial_z p)(\partial_x z)_s$ leading to

$$(\partial_x p)_z (\partial_z p)^{-1} = (\partial_z p)^{-1}(\partial_x p)_s - (\partial_x z)_s$$

In the particular case of the IFS coordinate $s$ we have

$$(\partial_z p)^{-1}(\partial_x p)_s = \frac{B(z)\partial_x p_s}{(\partial_z A) + (\partial_z B)p_s} \tag{B4}$$

and $(\partial_x z)_s$ can be estimated after integrating the hydrostatic balance.

Starting from the layer interfaces height $z^{\text{ifs}}_{i,j,k+1/2}$, surface pressure $(p_s)_{i,j}$ and parameter values $A_k, B_k, A_{k+1/2}, B_{k+1/2}$ the different steps are the following:

1. Compute $\Delta x_{i,j}$ and $\Delta y_{i,j}$ from latitudes and longitudes

2. Compute horizontal gradients $\partial_x p_s$ and $\partial_y p_s$ for surface pressure

$$\text{FX}_{i+1/2,j} = \frac{2\{(p_s)_{i+1,j} - (p_s)_{i,j}\}}{\Delta x_{i,j} + \Delta x_{i+1,j}}, \qquad \text{FY}_{i,j+1/2} = \frac{2\{(p_s)_{i,j+1} - (p_s)_{i,j}\}}{\Delta y_{i,j} + \Delta y_{i,j+1}}$$

3. Compute horizontal gradients $(\partial_x z)_s$ and $(\partial_y z)_s$

$$\text{dZx}_{i+1/2,j,k} = \frac{z^{\text{cep}}_{i+1,j,k+1/2} - z^{\text{cep}}_{i,j,k+1/2} + z^{\text{cep}}_{i+1,j,k-1/2} - z^{\text{cep}}_{i,j,k-1/2}}{\Delta x_{i,j} + \Delta x_{i+1,j}}$$

$$\text{dZy}_{i,j+1/2,k} = \frac{z^{\text{cep}}_{i,j+1,k+1/2} - z^{\text{cep}}_{i,j,k+1/2} + z^{\text{cep}}_{i,j+1,k-1/2} - z^{\text{cep}}_{i,j,k-1/2}}{\Delta y_{i,j} + \Delta y_{i,j+1}}$$

4. Compute $(\partial_z p)^{-1}(\partial_x p)_s$ via (B4)

$$\text{wrkX}_{i,j,k} = \frac{1}{2}\frac{B_k\left(z^{\text{cep}}_{i,j,k+1/2} - z^{\text{cep}}_{i,j,k-1/2}\right)\left(\text{FX}_{i+1/2,j} + \text{FX}_{i-1/2,j}\right)}{(p_s)_{i,j}(B_{k+1/2} - B_{k-1/2}) + (A_{k+1/2} - A_{k-1/2})}$$

$$\text{wrkY}_{i,j,k} = \frac{1}{2}\frac{B_k\left(z^{\text{cep}}_{i,j,k+1/2} - z^{\text{cep}}_{i,j,k-1/2}\right)\left(\text{FY}_{i+1/2,j} + \text{FY}_{i-1/2,j}\right)}{(p_s)_{i,j}(B_{k+1/2} - B_{k-1/2}) + (A_{k+1/2} - A_{k-1/2})}$$

5. Finalize (we get a minus sign in $R^u_{\text{LS}}$ because the grid in the $y$-direction is flipped in the raw data)

$$(R^u_{\text{LS}})_{i,j,k} = -g\left(\text{wrkY}_{i,j,k} - \frac{1}{2}(\text{dZy}_{i,j+1/2,k} + \text{dZy}_{i,j-1/2,k})\right)$$

$$(R^v_{\text{LS}})_{i,j,k} = -g\left(\text{wrkX}_{i,j,k} - \frac{1}{2}(\text{dZx}_{i+1/2,j,k} + \text{dZx}_{i-1/2,j,k})\right)$$

## Appendix C: Discrete algorithms to compute $l_{\text{up}}$ and $l_{\text{dwn}}$

In the following we describe the discrete algorithms used to provide the mixing lengths $l_{\text{up}}$ and $l_{\text{dwn}}$ given in Sect. 3.2.2. Four different ways to compute those quantities have been implemented in the ABL1d model.

## C1 Bougeault and Lacarrère (1989) length scale

The Bougeault and Lacarrère (1989) mixing length defines $l_{\mathrm{up}}$ and $l_{\mathrm{dwn}}$ as

$$\int\limits_{z}^{z+l_{\mathrm{up}}} N^2(s)(s-z)ds = e(z), \qquad \int\limits_{z-l_{\mathrm{dwn}}}^{z} N^2(s)(z-s)ds = e(z). \tag{C1}$$

By construction such mixing lengths are bounded by the distance to the bottom and the top of the computational domain and revert to the Deardorff (1980) length scale (i.e. $l_{\mathrm{up}} = l_{\mathrm{dwn}} = \sqrt{2e(z)/N^2}$) for $N^2 = $ cste. An objective is to satisfy this last property also at a discrete level. Considering a simple trapezoidal rule to approximate the integral in (C1) over each grid cells, the procedure for the computation of $l_{\mathrm{up}}(z_{k+1/2})$ is given in Algorithm 1. In the case $N^2(z_{p+1/2}) = N^2(z_{p-1/2}) = N_{\mathrm{cst}}^2$ ($\forall p$), Algorithm 1 gives the following sequence

$$\mathrm{FC}(z_{k+1/2}) = -e(z_{k+1/2})$$
$$\mathrm{FC}(z_{k+3/2}) = -e(z_{k+1/2}) + N_{\mathrm{cst}}^2 \frac{\mathrm{e3t}(z_{k+1})^2}{2}$$
$$\mathrm{FC}(z_{k+5/2}) = -e(z_{k+1/2}) + N_{\mathrm{cst}}^2 \frac{[\mathrm{e3t}(z_{k+1}) + \mathrm{e3t}(z_{k+2})]^2}{2}$$
$$\cdots \qquad \cdots$$

As soon as $\mathrm{FC}(z_{p+1/2})$ changes sign we stop the procedure because $l_{\mathrm{up}}$ such that $-e(z_{k+1/2}) + N_{\mathrm{cst}}^2 l_{\mathrm{up}}^2 = 0$, which corresponds
to the Deardorff (1980) length scale, has been found. We note $l_{\mathrm{BL89}}$ the mixing length corresponding to the Bougeault and Lacarrère (1989) algorithm.

---

**Algorithm 1** Procedure to compute the Bougeault and Lacarrère (1989) length scale $l_{\mathrm{up}}(z_{k+1/2})$.

---

Initialize FC : $\mathrm{FC}(z_{k+1/2}) = -e(z_{k+1/2})$

**for** $p = k+1, N$ **do**

$\quad \mathrm{FC}(z_{p+1/2}) = \mathrm{FC}(z_{p-1/2}) + \dfrac{\mathrm{e3t}(z_p)}{2}\left(N^2(z_{p+1/2})(z_{p+1/2} - z_{k+1/2}) + N^2(z_{p-1/2})(z_{p-1/2} - z_{k+1/2})\right)$

$\quad$ **if** $\mathrm{FC}(z_{p+1/2}) \times \mathrm{FC}(z_{p-1/2}) \leq 0$ **then**

$\qquad z_{k+1/2} + l_{\mathrm{up}} \in [z_{p-1/2}, z_{p+1/2}]$

$\qquad$ **break**

$\quad$ **end if**

**end for**

Linearly interpolate FC between $z_{p-1/2}$ and $z_{p+1/2}$ to find $z_\star$ such that $\mathrm{FC}(z_\star) = 0$.

---

## C2 Adaptation of NEMO's length scale

The standard NEMO algorithm (Sec. 10.1.3 in Madec, 2012) is much easier to discretize. As a first step the Deardorff (1980) length scale $l_{D80}$ is computed at cell interfaces, such that

$$(l_{D80})_{k+1/2} = \max \left( \sqrt{\frac{2e_{k+1/2}}{\max(N^2, N_\varepsilon^2)}}, l_{\min} \right)$$

with $N_\varepsilon^2$ the minimum stratification allowed whose value is set to the smallest positive real computer value. The vertical gradients of $l_{D80}$ are then limited such that they stay smaller than the variations of height. This amounts to compute $l_{up}$ and $l_{dwn}$ as

$$(l_{up})_{k-1/2} = \min \left( (l_{up})_{k+1/2} + e3t_k, (l_{D80})_{k-1/2} \right) \tag{C2}$$

$$(l_{dwn})_{k+1/2} = \min \left( (l_{dwn})_{k-1/2} + e3t_k, (l_{D80})_{k+1/2} \right) \tag{C3}$$

with $e3t_k$ the thickness of vertical layer $k$ (Fig. 2). The resulting mixing length is simply referred to as $l_{D80}$.

## C3 Rodier et al. (2017) length scale

Recently, Rodier et al. (2017) proposed to add a shear related term to (C1) such that the definition of $l_{up}$ and $l_{dwn}$ becomes

$$\int_z^{z+l_{up}} \left[ N^2(s)(s-z) + c_0 \sqrt{e(s)} \|\partial_s \mathbf{u}_h\| \right] ds = e(z), \qquad \int_{z-l_{dwn}}^z \left[ N^2(s)(z-s) + c_0 \sqrt{e(s)} \|\partial_s \mathbf{u}_h\| \right] ds = e(z). \tag{C4}$$

where $c_0$ is a parameter whose value should be smaller than $\sqrt{C_m/c_\varepsilon}$. At a discrete level, the FC function in Algorithm 1 is replaced by

$$
\begin{aligned}
FC(z_{p+1/2}) = FC(z_{p-1/2}) \quad &+ \quad \frac{e3t(z_p)}{2} \left( N^2(z_{p+1/2})(z_{p+1/2} - z_{k+1/2}) + N^2(z_{p-1/2})(z_{p-1/2} - z_{k+1/2}) \right) \\
&+ \quad c_0 \frac{e3t(z_p)}{2} \left( \sqrt{e(z_{p+1/2})} \|\partial_z \mathbf{u}_h(z_{p+1/2})\| + \sqrt{e(z_{p-1/2})} \|\partial_z \mathbf{u}_h(z_{p-1/2})\| \right)
\end{aligned}
$$

This mixing length will be referred to as $l_{R17}$.

## 825 C4 A local buoyancy and shear-based length scale

For the sake of computational efficiency, we have derived a local version of the Rodier et al. (2017) length scale (noted $l_{D80}^\star$) which is original to the present paper

$$l_{D80}^\star(z) = \frac{2\sqrt{e(z)}}{c_0 \|\partial_z \mathbf{u}_h\| + \sqrt{c_0^2 \|\partial_z \mathbf{u}_h\|^2 + 2N^2(z)}}.$$

Once $l_{D80}^\star$ has been computed at cell interfaces $z = z_k + 1/2$ we apply the limitations (C2) and (C3) as in the NEMO algorithm.

## Appendix D: Code performance and profiling

To finalize our description of the implementation of the simplified atmospheric boundary layer model in NEMO, we assess in this appendix the computational efficiency of our approach. We compare the performance of two simulations: one with a coupling with the ABL1d model (with 50 vertical levels) which requires reading 3D atmospheric data in input files, and one with a standard ASL forcing strategy which necessitates reading only 2D atmospheric data. For the coupling with ABL1d, we consider the $l_{\mathrm{D80}}$ mixing length which gave robustly good results across the different numerical tests investigated earlier in the manuscript. The simulations are performed with NEMO version 4.0 for the ORCA025 configuration previously described on 128 cores (Intel(R) Xeon(R) E5 processors 2.6 GHz) compiled with ifort (v13.0.1) using the "$-\mathrm{i4} - \mathrm{r8} - \mathrm{O3} - \mathrm{fp} - \mathrm{model\ precise} - \mathrm{fno}$ options. The I/Os are handled via Lustre file system. Each MPI subdomain has $80 \times 130$ points in the horizontal and 75 points in the vertical. The various reports given below have been obtained from a built-in NEMO code instrumentation dedicated to calculation measurement (e.g. Maisonnave and Masson, 2019). As mentioned earlier, the outputs are done using the parallel I/O capabilities provided by the XIOS library. Thanks to XIOS, we do not expect any significant difference between the two simulations regarding the cost of output operations. However, the use of XIOS to handle input operations is still under development and because of the significant amount of data to read in the ABL coupling strategy, it makes sense to assess the associated overhead. We ran the ASL forced and ABL coupled NEMO simulations for 20 days such that the cost of the initialization step is no longer visible in the averaged cost per time-step. Moreover, for the two simulations, the atmospheric data necessary for the computation of the turbulent components of air-sea fluxes are provided every 6 hours.

We first show in Fig. D1 the elapsed time for each time-step over the first 48 h of the simulations with different ways to specify the surface fluxes. For most time-steps, the overhead associated with the ABL1d when using the $l_{\mathrm{D80}}$ mixing length is very small (in the order of $4\%$), however every 18 time-steps (i.e. every 6 hours), there is a larger overhead due to the input part of the I/O operations. To further refine our assessment, we report in Tab. D1 the elapsed and CPU time spent on average over all the processors in the 11 most expansive sections of the code. As expected, the CPU time is not significantly affected by the ABL1d model (increase of $4\%$), but the elapsed time is increased by about $9\%$ because of the time spent in waiting for I/O operations.

The overhead associated with input operations could be mitigated by reducing the number of vertical levels in the ABL1d model (we used 50 levels here to get an upper bound on the computational overhead) and either by using XIOS to handle input operations or by running ABL1d in detached mode as explained in Sec. 3.4 such that the time spent reading input files is covered by actual computations. Nonetheless the small increase in CPU time leaves room for further improvements of the ABL model to relax the horizontal homogeneity assumption.

*Author contributions.* Florian Lemarié has written the paper with the help of all the coauthors. Florian Lemarié, Guillaume Samson, and Gurvan Madec designed and developed a preliminary version of the ABL1d model within NEMO 3.6 stable version. This original code was then ported to NEMO release 4.0. Jean-Luc Redelsperger and Hervé Giordani have provided inputs in the design of the TKE closure scheme

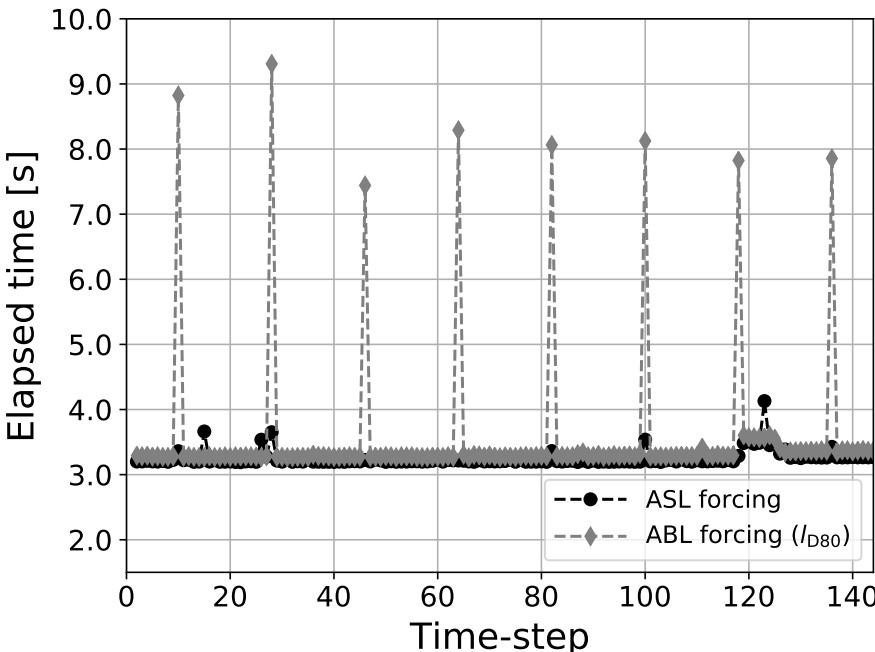

**Figure D1.** Elapsed time for each time-step of a 48 h simulation with standard ASL forcing strategy (black circles) and ABL forcing strategy using the $l_{D80}$ mixing length (grey diamonds). For the two simulations the time-step size is $\Delta t = 1200$ s.

and of the numerical experiments. Florian Lemarié has carried out the idealized numerical experiments, Guillaume Samson the realistic experiments and Jean-Luc Redelsperger the MesoNH simulations.

*Competing interests.* The authors declare that they have no conflict of interest.

*Disclaimer.* TEXT

*Acknowledgements.* We thank Anton Beljaars and one anonymous reviewer whose efforts helped to improve earlier versions of this paper. We also thank Pascal Marquet and Sébastien Masson for useful discussions. Florian Lemarié, Guillaume Samson, and Gurvan Madec have received funding from the European Union's Horizon 2020 research and innovation programme under grant agreement No 821926 (IMMERSE). Florian Lemarié and Jean-Luc Redelsperger also acknowledge the support by Mercator-Ocean and the Copernicus Marine
Environment Monitoring Service (CMEMS) through contract 22-GLO-HR - Lot 2 (High-resolution ocean, waves, atmosphere interaction).

| Section | ASL forcing | | ABL coupling | | Description |
|---|---|---|---|---|---|
| | Elapsed | CPU | Elapsed | CPU | |
| dyn_spg | 789.2 s (15.22%) | 786.5 s (15.42%) | 778.2 s (13.79%) | 775.5 s (14.62%) | Non-linear free surface |
| icedyn_rhg | 637.9 s (12.30%) | 639.2 s (12.53%) | 634.7 s (11.24%) | 638.6 s (12.04%) | Sea-ice rheology |
| tra_adv | 623.9 s (12.03%) | 613.8 s (12.04%) | 626.0 s (11.09%) | 615.9 s (11.61%) | 3D tracer advection with FCT scheme |
| zdf_phy | 546.6 s (10.54%) | 545.4 s (10.70%) | 541.0 s (9.59%) | 538.8 s (10.16%) | Vertical physics: surface boundary layer + internal wave mixing |
| dyn_adv | 229.1 s (4.42%) | 227.9 s (4.47%) | 229.2 s (4.06%) | 227.9 s (4.30%) | 3D Nonlinear momentum advection |
| tra_ldf | 221.0 s (4.26%) | 220.0 s (4.31%) | 220.3 s (3.90%) | 219.4 s (4.14%) | Isoneutral diffusion operator |
| ldf_slp | 185.5 s (3.58%) | 184.2 s (3.61%) | 186.6 s (3.30%) | 184.35 s (3.48%) | Computation of local neutral directions |
| dom_vvl | 245.4 s (4.74%) | 229.7 s (4.51%) | 243.6 s (4.32%) | 228.5 s (4.31%) | Lagrangian evolution of vertical scale factors with free surface |
| dyn_nxt | 159.9 s (3.08%) | 151.5 s (2.97%) | 159.0 s (2.82%) | 150.8 s (2.84%) | barotropic/baroclinic coupling and Asselin time filtering |
| dyn_zdf | 131.8 s (2.54%) | 131.3 s (2.57%) | 130.8 s (2.32%) | 130.3 s (2.46%) | Apply bottom and surface stress and solve the implicit vertical mixing |
| **sbc** | **101**.6 s (1.96%) | **92.77** s (1.82%) | **580**.8 s (10.29%) | **327**.4 s (6.17%) | Surface flux computation (turbulent and non-turbulent) |
| … | … | … | … | … | |
| Total | 5184.985 s | 5099.884 s | 5643.897 s | 5302.949 s | |

**Table D1.** Report of the elapsed time and CPU time in different sections of the NEMO(v4.0) code for the ASL forcing strategy (left portion of the table) and the ABL coupling strategy (right portion of the table). The timing is averaged on all processors. The right most column provides a quick description of the task handled by the corresponding section. On top of the timing in seconds the percentage of the total CPU and elapsed associated to each section is reported in parentheses. The computational overhead associated to the ABL coupling strategy can be estimated from the sbc section and the elapsed/CPU time.

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
