# Peer review of "A simplified atmospheric boundary layer model for an improved representation of air-sea interactions in eddying oceanic models: implementation and first evaluation in NEMO(4.0)"

_Geoscientific Model Development, 2020_

## Referee Comment (RC1) · Anton Beljaars (Referee) · 29 Aug 2020

This paper proposes a point wise single column approach to include feedbacks of ocean SST and current on near surface atmospheric variables in offline ocean simulations with prescribed atmospheric conditions e.g. from re-analysis. This is obviously of interest for the development of high resolution ocean models, because fully coupled simulations are expensive and running fully coupled at very high resolution, e.g. at 1km, is still not feasible at the global scale. The paper documents the infrastructure that has been built for this purpose, with elements like boundary layer parametrization,

numerical techniques, and technical aspects. The evaluation of the boundary layer scheme on the basis of LES is comprehensive and in fact impressive. It clearly shows that some versions of the closure are better than others, but also that advection is important in case of SST gradients. Although important, the handling of advection is left to future work.

Also first experiments with offline ocean simulations are presented with prescribed atmospheric forcing. Simulations with traditional forcing at 10m are compared with simulations that are forced and interactive with data over the entire boundary layer. Evaluation is limited to a qualitative comparison of correlation between wind and SST, and the relation between surface current vorticity and wind or stress curl. However, I suspect that the results on these coupling coefficients depend on the strength of relaxation coefficients. A clean and objective optimization strategy for the relaxation is not obvious and left to future work.

The paper covers a lot of ground and as a purely scientific paper, more would be needed on evaluation, comparison with observations and optimization of the relaxation. On the other hand, I very much welcome this paper as a step towards a technical infrastructure for offline ocean simulations with realistic air-sea interaction. This is very much needed not only for offline ocean simulations, but also for coupled simulations with a lower resolution atmosphere. In the latter case, the air-sea interaction at high resolution can be improved by the type of scheme that is proposed here. Furthermore, I expect these type of intermediate complexity systems to play an important role in coupled data assimilation, which is hard to do in fully coupled models. The merit of this paper is that it carefully describes the design and evaluation of a technical infrastructure that can be used in further studies. GMD is highly suitable for this type of paper, so I recommend publication, after addressing the points below.

Lines 61-71 The authors motivate the need for a comprehensive boundary layer feedback in the ocean coupling by making reference to earlier studies, which is good. Unfortunately, this paragraph is hard to read, mainly because too many aspects are put

together here. It is perhaps better not to discuss currents at this point because the effects of currents can (or is) already included in ASL coupling. Also the reference to bottom drag does not help. The main point is that with boundary layer coupling, temperature and wind at 10m change when heat and momentum fluxes change.

Section 2.2 This section presents the basic idea, which is central to the paper. Ideally, one would like to have the full set of atmospheric equations to evolve the column and add the nudging term only to keep the forcing "deterministic" or reproducible. The purpose of the term is to ensure that the chaotic atmosphere does not drift off and one would like the nudging term as small as possible. As soon as the atmospheric equations are less complete, the nudging term has to work harder. The question is how accurate is the selected single column approximation? Is it possible to motivate the approximations by an asymptotic framework like in the surface layer? It would be good to say something about the relative magnitude of different terms: temporal change, advection, pressure gradient, Coriolis, and diffusion dependent on the traditional Rossby number and dependent on a dimensionless number that describes the magnitude of the diffusion term relative to the advection or pressure gradient terms. In this paper the diffusion term is considered to be the dominant term, but for the momentum equation Coriolis and pressure gradient are added. This is probably quite good for large horizontal scales, but becomes questionable for the 1km scale. The question can also be asked in a different way, namely what is the equilibrium solution of the diffusion equation with boundary conditions at the surface and just above the boundary layer? Over the ocean a steady state solution can be very accurate provide that all the forcing terms are present, namely radiative flux divergence, Coriolis and pressure gradient. The moisture equation is even simpler; without cloud and precipitation processes, only diffusion is left and for steady state there is no flux divergence. In conclusion, I think that the proposed approach describes the effects of an instantaneous response of the entire boundary layer to changes in the surface boundary condition.

Section 2.4 and line 242 Relaxation is an important ingredient of the proposed system

and has a strong influence as suggested at line 235. It was decided to scale the relaxation time scale with the model time step. This is understandable for the top of the boundary layer where relaxation is used to impose a boundary condition by relaxing to the forcing in a single time step (immersed boundary condition). However, in the boundary layer, I would have expected a more "physical" time scale, dependent on which physical process it represents, or how fast the error growth is in the chaotic atmosphere.

Section 3.3 If I understand correctly, the coupling with sea ice involves the averaging of temperatures of different categories before computing fraction weighted fluxes over open water and sea ice (lines 384-385). Alternatively, it would have been possible to extend the weighted averaging in 384-385 to all the categories (as e.g. in Best et al., 2004, J. Hydrometeor., 5, 1271-1278 for land use categories). It is not the same as averaging the sea ice temperature, because the transfer coefficients are stability dependent.

Section 5.2 Wind to SST correlation is considered here. I have seen papers by Chelton, where the emphasis is on wind to SST-gradient correlation. Is the latter correlation less realistic in the current coupling because advection is missing? Figures 10 and 11 are discussed and compared with figures in literature. The paper is already (too) long, but it should be possible to read the paper on its own, so it might be better to reduce the number of plots from 3 to 2 and add the reference figures.

Section 5.3 This section (including Table 4) describes the computer technology aspects, but I feel that it is a bit too technical for this paper. I recommend to shorten it and limit to general results on processing and IO time.

Section 6 This section summarizes the paper, and describes plans for future work. Also the inclusion of advection is mentioned. I suggest to conclude more explicitly that advection is high priority. From the experiments with an SST front it is clear that advection can not be ignored at high resolution.

---

## Referee Comment (RC2) · Anonymous Referee #2 · 29 Sep 2020

First, I would like to commend the authors for tackling this much-needed task. When forced by prescribed atmospheric fields via bulk formula, ocean-ice only models can only modify the atmospheric forcing via the drag coefficient. In a fully coupled system, the atmosphere is, in turn, expected to respond the ocean SST and currents. There is therefore a strong need for intermediate models like the proposed ABL1d by allowing part of the near surface atmospheric field to vary as a function of ocean variables.

Second, while the authors provide a thorough description of their approach, it is often dense and not always easy to extract the main information. It is also lacking an

overview of what is the current practice in planetary boundary layer models and how the particular approach chosen for ABL1d stacks against them.

Major comment #1: While I appreciate the fact that this approach emphasize over water conditions, the paper would benefit from a brief overview of current PBL and parameterizations (Baklanov et al., BAMS, 2011, DOI:10.1175/2010BAMS2797.1) and why the authors decided to use their own approach. In particular, there are already existing standalone PBL models such as the one from the University of Washington. Important differences between the planetary boundary layer over land and ocean surfaces arise because the ocean thermodynamic and dynamic characteristics, especially its temperature and this should be contrasted with existing models. This would set the stage for section 2.3

Major comment #2: It is somewhat related to #1, but when stating that the turbulent mixing by the air-sea feedback is thought to be the main coupling mechanism and that this mechanism is expected to explain most of the eddy-scale wind-SST and wind-currents interactions, this needs to be further substantiated or be made clear that this is one of your assumptions.

Major comment #3: The series of validation experiments in sections 4 and 5 are not easy to ready and would benefit from a clearer introduction clearly stating which aspect of the ABL1d model is being tested and which limitations are emphasized. A thorough discussion of the choices in relaxation time scales and lack of advection are key elements to the validation discussion. Section 5.2 is the main achievement with an application of global NEMO, but what are we learning here besides the fact that it has impact on the circulation? For each of the applications/validations sections, the manuscript would benefit from an introductory statement describing the intent of each section, what is being tested, and their outcome.

Minor comment: I suggest moving the code performance section to an appendix.

Recommendation: Accept after revisions – minor in the sense that they do not require

new experiments, major in the sense that the paper would benefit from a substantial rewrite with a stronger introduction with a discussion of current PBL practices and another one clearly describing the motivations for the validation/applications.

---

## Author Response (AR1)

**Response to reviewers on manuscript gmd-2020-210**

First, the authors would like to sincerely thank the reviewers for their careful reading of the paper and their valuable comments to the manuscript and helpful suggestions. We further clarified several issues raised during the review process. Please find attached our revised paper and below a summary of how we responded to the comments. We also provide a diff marked-up file. Responses to the specific concerns raised by both reviewers are reported in blue color below.

During the revision process it became clear to us that our objective to make this paper readable by a large audience including model developers, oceanic model users, as well as the OA coupling and PBL modeling communities while providing enough information to make our results easily reproducible is ambitious. There is a risk to overwhelm the reader with too much information. When preparing the revised manuscript we tried to improve the readability through the following changes:

1. The introduction has been reshaped with subsections to clearly separate the different motivations and also highlight the assumptions inherent to our approach.

2. Two technical sections of the paper (the discrete algorithms to compute mixing lengths and the profiling of the NEMO code) have been moved to the appendixes.

3. Section 4 in the original manuscript was hard to follow because we did not state clearly what were our objectives when running a particular numerical simulation. We have added a new subsection (Subsec. 4.1) to help the reader and explicitly give our motivations.

With those modifications we managed to reduce the size of the main body of the paper by 2 pages.

**1 Response to reviewer #1 (Anton Beljaars)**

*This paper proposes a point wise single column approach to include feedbacks of ocean SST and current on near surface atmospheric variables in offline ocean simulations with prescribed atmospheric conditions e.g. from re-analysis. This is obviously of interest for the development of high resolution ocean models, because fully coupled simulations are expensive and running fully coupled at very high resolution, e.g. at 1km, is still not feasible at the global scale. The paper documents the infrastructure that has been built for this purpose, with elements like boundary layer parametrization, numerical techniques, and technical aspects. The evaluation of the boundary layer scheme on the basis of LES is comprehensive and in fact impressive. It clearly shows that some versions of the closure are better than others, but also that advection is important in case of SST gradients. Although important, the handling of advection is left to future work. Also first experiments with offline ocean simulations are presented with prescribed atmospheric forcing. Simulations with traditional forcing at 10m are compared with simulations that are forced and interactive with data over the entire boundary layer. Evaluation is limited to a qualitative comparison of correlation between wind and SST, and the relation between surface current vorticity and wind or stress curl. However, I suspect that the results on these coupling coefficients depend on the strength of relaxation coefficients. A clean and objective optimization strategy for the relaxation is not obvious and left to future work. The paper covers a lot of ground and as a purely scientific paper, more would be needed on evaluation, comparison with observations and optimization of the relaxation. On the other*

*hand, I very much welcome this paper as a step towards a technical infrastructure for offline ocean simulations with realistic air-sea interaction. This is very much needed not only for offline ocean simulations, but also for coupled simulations with a lower resolution atmosphere. In the latter case, the air-sea interaction at high resolution can be improved by the type of scheme that is proposed here. Furthermore, I expect these type of intermediate complexity systems to play an important role in coupled data assimilation, which is hard to do in fully coupled models. The merit of this paper is that it carefully describes the design and evaluation of a technical infrastructure that can be used in further studies. GMD is highly suitable for this type of paper, so I recommend publication, after addressing the points below.*

- **Lines 61-71**: The authors motivate the need for a comprehensive boundary layer feed- back in the ocean coupling by making reference to earlier studies, which is good. Un- fortunately, this paragraph is hard to read, mainly because too many aspects are put together here. It is perhaps better not to discuss currents at this point because the effects of currents can (or is) already included in ASL coupling. Also the reference to bottom drag does not help. The main point is that with boundary layer coupling, temperature and wind at 10m change when heat and momentum fluxes change.

  Yes we agree that this part of the introduction was hard to follow. The introduction is now structured with subsections and subsection 1.2 specifically addresses the motivations in terms of processes. Within this subsection we clearly separate the 3 mechanisms found in the literature to explain air-sea interactions at eddy-scale (i.e. the downward momentum mixing, the pressure adjustment and the current feedback). It is indeed true that the effect of the currents is also present with the ASL coupling strategy but the effect is largely over-estimated. From our point of view model users should avoid taking into account oceanic currents with the ASL forcing strategy because the ocean would lose too much energy. Correcting this over-estimation problem for uncoupled simulations was a strong motivation at the beginning of our work, therefore we decided to keep this aspect in the introduction but hopefully things are more clearly explained in the revised manuscript.

- **Section 2.2**: This section presents the basic idea, which is central to the paper. Ideally, one would like to have the full set of atmospheric equations to evolve the column and add the nudging term only to keep the forcing "deterministic" or reproducible. The purpose of the term is to ensure that the chaotic atmosphere does not drift off and one would like the nudging term as small as possible. As soon as the atmospheric equations are less complete, the nudging term has to work harder. The question is how accurate is the selected single column approximation? Is it possible to motivate the approximations by an asymptotic framework like in the surface layer? It would be good to say something about the relative magnitude of different terms: temporal change, advection, pressure gradient, Coriolis, and diffusion dependent on the traditional Rossby number and dependent on a dimensionless number that describes the magnitude of the diffusion term relative to the advection or pressure gradient terms. In this paper the diffusion term is considered to be the dominant term, but for the momentum equa- tion Coriolis and pressure gradient are added. This is probably quite good for large horizontal scales, but becomes questionable for the 1km scale. The question can also be asked in a different way, namely what is the equilibrium solution of the diffusion equation with boundary conditions at the surface and just above the boundary layer? Over the ocean a steady state solution can be very accurate provide that all the forcing terms are present, namely radiative flux divergence, Coriolis and pressure gradient. The moisture equation is

even simpler; without cloud and precipitation processes, only diffusion is left and for steady state there is no flux divergence. In conclusion, I think that the proposed approach describes the effects of an instantaneous response of the entire boundary layer to changes in the surface boundary condition.

*We agree that the question you raise : how accurate is the selected single column approximation? is fundamental for the future evolution of our approach. Several authors have already shown that momentum vertical turbulent mixing, pressure gradient, Coriolis, and nonlinear advection are all important to the momentum balance in the marine atmospheric boundary layer at the vicinity of oceanic fronts (see for example Spall (2007), Small et al. (2008) or ONeill et al. (2010) ). It is also well known that the relative importance of those terms depends on the wind regime: for strong winds the vertical mixing is the dominant mechanism while for weak winds the pressure adjustment mechanism is dominant. We did not provide a detailled discussion on this aspect in the original version of the manuscript because the current single-column approximation is only a first step before testing more complex formulations. Our rationale is that the two main bottlenecks in term of computational cost inherent to the ABL forcing strategy are the reading of 3D atmospheric data and the ABL turbulent scheme. As a first step, we focused on those two aspects to assess whether or not our approach would be a viable option (see it as a proof of concept). Because turbulent schemes are usually tested in a single-column setup we decided to start with such formulation. At least it allows for a representation of the modification of atmospheric stability due to SST (surface wind increases over warm water and decreases over cold water) which already brings an improvement compared to the ASL forcing strategy. Now that we have an adequate computational framework and an efficient turbulent scheme that we can operate for a reasonable overhead in term of computational cost we can move from the development phase to the next phase. Even if we go for more complicated formulations of our simplified model the steps described in the paper are necessary steps. We now make it more clear in the introduction (see subsection 1.3) and the conclusion of the paper.*
*Thanks for your suggestions, we will consider them as we now started to go forward on more advanced formulation of our model. We believe that a promising way to include horizontal advection and to get rid of the tricky nudging term is to formulate the ABL model as a perturbation around a time-evolving ambient state provided by data from a large-scale atmospheric model. This is very much inline with the notion of soundproof equations (see papers by D. Durran, A. Arakawa or P. Smolarkiewicz). This is an ongoing work and we will keep you informed of our findings.*

- **Section 2.4 and line 242**: Relaxation is an important ingredient of the proposed system and has a strong influence as suggested at line 235. It was decided to scale the relaxation time scale with the model time step. This is understandable for the top of the boundary layer where relaxation is used to impose a boundary condition by relaxing to the forcing in a single time step (immersed boundary condition). However, in the boundary layer, I would have expected a more "physical" time scale, dependent on which physical process it represents, or how fast the error growth is in the chaotic atmosphere.

*We agree with this remark, and on top of that the recommendation given in the previous version of the manuscript (i.e. choose $\lambda_s^{\min}$ such that $\lambda_s^{\min}\Delta t \approx 0.1$) was not consistent with the parameter values used for the numerical experiments in Sect. 5. This paragraph has been reformulated in the new version of Sect. 2.4. and we now consider a typical adjustment timescale of the ABL to surface perturbations to define the relaxation time-scale in the lower*

portion of the boundary layer. We believe it is now clear that the $\lambda_s^{\min}$ parameter should not change when changing the time-step of the model because it is a physical parameter and not a numerical one, only $\lambda_s^{\max}$ has to change with $\Delta t$. For the realistic simulations we considered a relaxation time-scale around 90 min which roughly corresponds the time it takes for a parcel with vertical velocity $W \sim 0.5\mathrm{m\ s}^{-1}$ to reach the top of the PBL (with $H_{\mathrm{pbl}} \approx 1300\mathrm{m}$) and come back to the surface.

- **Section 3.3**: If I understand correctly, the coupling with sea ice involves the averaging of temperatures of different categories before computing fraction weighted fluxes over open water and sea ice (lines 384-385). Alternatively, it would have been possible to extend the weighted averaging in 384-385 to all the categories (as e.g. in Best et al., 2004, J. Hydrometeor., 5, 1271-1278 for land use categories). It is not the same as averaging the sea ice temperature, because the transfer coefficients are stability dependent.

  This is a very good point, thanks for raising it. In our case the two options you mention are identical because in the current version of NEMO, all sea-ice categories have the same transfer coefficients $C_d^{\mathrm{ice}}$, $C_h^{\mathrm{ice}}$ and $C_e^{\mathrm{ice}}$ and the same sea-ice velocities, meaning that

  $$C_h^{\mathrm{ice}}\|\mathbf{u}^{\mathrm{atm}} - \mathbf{u}^{\mathrm{ice}}\|\left(T^{\mathrm{atm}} - \sum_l a_l T_l^{\mathrm{ice}}\right) = \sum_l a_l C_h^{\mathrm{ice}}\|\mathbf{u}^{\mathrm{atm}} - \mathbf{u}^{\mathrm{ice}}\|\left(T^{\mathrm{atm}} - T_l^{\mathrm{ice}}\right)$$

  But we fully agree that in the more general case where $C_h^{\mathrm{ice}}$ is function of $T_l^{\mathrm{ice}}$ averaging the fluxes over sea-ice categories will be different from averaging the temperatures before computing a single flux. We now mention this in the revised Section 3.3 because there is already a plan to have a more advanced bulk formulation over sea-ice in NEMO. An other option we have considered would be to run the whole atmospheric column over each sea-ice categories and then average over those different columns. This would give a different result from the two other options but for a much larger computational cost.

- **Section 5.2** Wind to SST correlation is considered here. I have seen papers by Chelton, where the emphasis is on wind to SST-gradient correlation. Is the latter correlation less realistic in the current coupling because advection is missing?

  This is an interesting remark. We chose to show the wind to SST correlation to allow a direct comparison with the spatial map given in Bryan et al. (2010). In particular Bryan et al. (2010) show that the correlation between SST and surface wind-speed depends strongly on the horizontal resolution (their Fig. 1 illustrates this aspect). The ability to reproduce the modification of surface wind field by mesoscale SST is a prerequisite to be able to represent the resulting convergences and divergences of the surface winds. We believe that Chelton and co-authors have mostly looked at SST induced changes in wind-stress rather than surface winds (but it is probably safe to consider that changes in wind stress are mostly attributable to changes in surface wind speed). They came up with

  $$\nabla \times \boldsymbol{\tau} = c_1 \nabla \mathrm{SST} \times \widehat{\boldsymbol{\tau}}$$
  $$\nabla \cdot \boldsymbol{\tau} = c_2 \nabla \mathrm{SST} \cdot \widehat{\boldsymbol{\tau}}$$

  with $\widehat{\boldsymbol{\tau}}$ the unit vector in the wind-stress direction. Following your comment, we have computed the correlation factor between $\nabla \mathrm{SST} \cdot \widehat{\boldsymbol{\tau}}$ and $\nabla \cdot \boldsymbol{\tau}$ for our NEMO/ABL1d ORCA025 simulation and we compare it to a coupled WRF-NEMO simulation provided by Lionel Renault. This comparison is shown in Fig. 1. It seems that our results compare well with the ones from coupled simulations.

$$\text{Cor}\left(\nabla\text{SST}\cdot\hat{\boldsymbol{\tau}},\nabla\cdot\boldsymbol{\tau}\right)$$

[Figure]

Figure 1: Correlation coefficient between downwind SST gradients and surface wind-stress divergence.

- **Figures 10 and 11** are discussed and compared with figures in literature. The paper is already (too) long, but it should be possible to read the paper on its own, so it might be better to reduce the number of plots from 3 to 2 and add the reference figures.

  We merged figures 10 & 11 in one figure (Fig. 10 in the revised manuscript). If we understood correctly you suggest to include in our paper the figures from Bryan et al. (2010) and Renault et al. (2019) we used to assess our results ? We include those figures at the end of this document (see App. A) but we are not sure that it is possible to integrate those figures directly into our manuscript... Maybe we misunderstood your suggestion.

- **Section 5.3** This section (including Table 4) describes the computer technology aspects, but I feel that it is a bit too technical for this paper. I recommend to shorten it and limit to general results on processing and IO time.

  We decided to move this section (Sect. 5.3 in the previous version of the manuscript) to appendix D and to discuss the important outcomes in the conclusion.

- **Section 6** This section summarizes the paper, and describes plans for future work. Also the inclusion of advection is mentioned. I suggest to conclude more explicitly that advection is high priority. From the experiments with an SST front it is clear that advection can not be ignored at high resolution.

  Yes this is more clearly stated in the revised conclusion. As mentioned earlier, our next step forward now that we have a reliable turbulent scheme and an adequate computational framework is to study different level of complexity in the formulation of the simplified ABL model.

**Response to referee # 2**

*First, I would like to commend the authors for tackling this much-needed task. When forced by prescribed atmospheric fields via bulk formula, ocean-ice only models can only modify the atmospheric forcing via the drag coefficient. In a fully coupled system, the atmosphere is, in turn, expected to respond the ocean SST and currents. There is therefore a strong need for intermediate models like the proposed ABL1d by allowing part of the near surface atmospheric field to vary as a function of ocean variables.*

Thanks for those encouraging remarks

*Second, while the authors provide a thorough description of their approach, it is of- ten dense and not always easy to extract the main information.*

Yes we agree that the paper is dense, therefore we tried to improve its readability. As mentioned in the preamble of our reply, we have moved to appendices two technical subsections of the paper and in Section 4 we now provide the main outcomes for each of the numerical experiments we conduct.

*It is also lacking an overview of what is the current practice in planetary boundary layer models and how the particular approach chosen for ABL1d stacks against them.*

We indeed do not provide an overview of current practices in PBL models (beyond citing Baklanov et al. 2011 & LeMone et al., 2019 in the revised manuscript) because for our purposes we were looking for a simple (i.e. without cloud-related processes) and efficient scheme with a limited number of arbitrary parameters to set. Our objectives are different from the current concerns in the PBL community which is more focused on cloud processes, flows over complex topography and structural aspects like the addition of a Turbulent Potential Energy equation or mass-flux schemes. We raise this point in the revised subsection 1.3. For the particular PBL scheme we use (a prognostic TKE with diagnostic mixing length turbulent scheme) most of the research is oriented toward an appropriate choice of mixing length, that's the reason why we investigate the sensitivity to this aspect.

In regard to your remark on the lack of comparison with existing PBL models. Note that the testcases we discuss in sections 4.2, 4.3, 4.4.1, and 4.4.2 are standardized testcases routinely used in many publications dealing with turbulence modelling of the atmospheric boundary layer under idealized forcing conditions. Our results can thus be directly confronted with the ones obtained with other PBL schemes. Comparison of results obtained with the present scheme to the ones used in published testcases show that the scheme is doing very well. Moreover, in the paper we show that our results compare favorably with the ones obtained from LES simulations using a well established model (MesoNH).

**Major comments:**

- **Major comment #1**: While I appreciate the fact that this approach emphasize over water conditions, the paper would benefit from a brief overview of current PBL and param- eterizations (Baklanov et al., BAMS, 2011, DOI:10.1175/2010BAMS2797.1) and why the authors decided to use their own approach. In particular, there are already existing standalone PBL models such as the one from the University of Washington. Important differences between the planetary boundary layer over land and ocean surfaces arise because the ocean thermodynamic and dynamic characteristics, especially its temperature and this should be contrasted

with existing models. This would set the stage for section 2.3

*We have thought a lot about the possibility to start with an existing PBL scheme (either standalone or extracted from an AGCM). But at some point it came clear to us that it would be much easier to take a fresh start. There are several reasons for that:*

- *Computational efficiency: each code is designed with particular choices of parallelization options (shared vs distributed memory), particular memory access patterns which can both dramatically affect the performance on massively parallel environments. Moreover developing a robust and efficient interface with external codes is not always an easy task. Because we are targeting an application in the operational context, those aspects are important.*

- *Code maintenance: having a turbulent scheme inside the NEMO framework and following the standard NEMO coding rules guarantees that the corresponding piece of code will be maintained and will go through automatic regression tests.*

- *Flexibility: unlike most atmospheric models our aim was to use a simple geopotential vertical coordinate. Moreover, we wanted to be able to easily investigate the sensitivities to model parameters and closure choices which has been straightforward in the framework we developed.*

*We were confident that this choice was the right one because some researchers involved in the project have a thorough expertise in prognostic TKE turbulent schemes for the atmospheric and oceanic PBL. In the end, our comparison with the MesoNH results shows that our scheme behaves very much like the original CBR-1d scheme used at Meteo France over the last two decades.*

*This is explained at the beginning of section 2.3 in the paper:*

*The turbulence scheme we have implemented in our ABL1d model is very similar to the so-called CBR-1d scheme of Cuxart et al. (2000) which is used operationally at Meteo France (Bazile et al., 2012). We chose to recode the parameterization from scratch for several reasons: computational efficiency, consistency with the NEMO coding rules, use of a geopotential vertical coordinate, and flexibility to add elements specific of the marine atmospheric boundary layer.*

- **Major comment #2**: It is somewhat related to #1, but when stating that the turbulent mixing by the air-sea feedback is thought to be the main coupling mechanism and that this mechanism is expected to explain most of the eddy-scale wind-SST and wind- currents interactions, this needs to be further substantiated or be made clear that this is one of your assumptions.

*Yes we agree with this remark. This assumption was clearly stated only in section 2.2 in the original manuscript. We now mention it explicitly in the introduction (see Sec. 1.3):*

*"Our aim is to account for the modulation of atmospheric turbulence by anomalies in sea-surface properties in the air-sea fluxes computation which is thought to be the main coupling mechanism at the characteristic scales of the oceanic mesoscales."*

*and in the conclusion:*

*"A crucial hypothesis is that the dominant process at the characteristic scale of the oceanic mesoscale is the so-called downward mixing process which stems from a modulation of atmospheric turbulence by sea surface temperature (SST) anomalies"*

*As said in our reply to reviewer #1, this assumption will be relaxed in future work. The main*

target here was to build the adequate computational framework within NEMO and to design an appropriate turbulence scheme.

- **Major comment #3**: The series of validation experiments in sections 4 and 5 are not easy to ready and would benefit from a clearer introduction clearly stating which aspect of the ABL1d model is being tested and which limitations are emphasized. A thorough discussion of the choices in relaxation time scales and lack of advection are key elements to the validation discussion. Section 5.2 is the main achievement with an application of global NEMO, but what are we learning here besides the fact that it has impact on the circulation? For each of the applications/validations sections, the manuscript would benefit from an introductory statement describing the intent of each section, what is being tested, and their outcome.

  This is a very good suggestion. We have added a subsection (subsection 4.1) to explain the objectives behind each numerical experiment. Then for each experiment we have summarized the important outcomes in the revised manuscript. The discussion on the appropriate choice of relaxation time-scale has been revised (see Sec. 2.4 and reply to reviewer # 1).
  Regarding your remark on Sect. 5.2., this paper should be primarily understood as a model development paper rather than a paper about the physics of the OA coupling. In section 5.2, a first lesson is that the NEMO model can be run efficiently, stably and without significant drift when coupled with ABL1d. Then, our second intent was to show that the various mechanisms mentioned in the introduction (namely the downward mixing process and the current feedback effect) are qualitatively and quantitatively well represented in the NEMO simulations with the ABL coupling strategy whereas they are absent in the ASL forcing strategy. We have split Section 5.2 in three subsections to clarify this point.
  Note that the Brivoal et al. (2020) paper is now available online (the reference has been updated in the revised manuscript). This paper is a process-oriented study of NEMO-ABL1d coupled model solutions.

**Minor comment:**

I suggest moving the code performance section to an appendix.

  Yes, reviewer#1 had the same request therefore we decided to move this section (Sect. 5.3 in the previous version of the manuscript) to appendix D and to discuss the important outcomes in the conclusion.

**A Reference figures from Bryan et al. (2010) and Renault et al. (2019)**

[Figure]

Figure 2: Fig. 1c from Bryan et al. (2010)

[Figure]

Figure 3: Fig. 1b from Renault et al. (2019)

[Figure]

Figure 4: Fig. 2c from Renault et al. (2019)

[revised manuscript text omitted]
_{\mathrm{sfc}}} = C_D \|\mathbf{u}_h(z_{\mathrm{sfc}}) - \mathbf{u}_{\mathrm{oce}}\|(\mathbf{u}_h(z_{\mathrm{sfc}}) - \mathbf{u}_{\mathrm{oce}}), \tag{3}$$

$$K_s \partial_z \phi|_{z=z_{\mathrm{sfc}}} = C_\phi \|\mathbf{u}_h(z_{\mathrm{sfc}}) - \mathbf{u}_{\mathrm{oce}}\|(\phi(z_{\mathrm{sfc}}) - \phi_{\mathrm{oce}}), \qquad \text{with } \phi = \theta, q \tag{4}$$

For the sake of consistency, it is preferable to use a bulk formulation as close as possible to the one used to compute the three-dimensional large-scale atmospheric data $\phi_{\mathrm{LS}}^{\mathrm{atm}}$. Because in the present study the plan is to use a large-scale forcing from ECMWF reanalysis products, we use the IFS[2] bulk formulation such as implemented in the AeroBulk[3] package (Brodeau et al., 2017) to compute $C_D$, $C_\theta$, and $C_q$ in realistic simulations (see  Sect. 5). Note that for an ASL forcing strategy $\mathbf{u}_h(z_{\mathrm{sfc}})$ and $\phi(z_{\mathrm{sfc}})$ in (3) would be respectively equal to $\mathbf{u}_{\mathrm{LS}}(z = 10\,\mathrm{m})$ and $\phi_{\mathrm{LS}}(z = 10\,\mathrm{m})$ while in the ABL coupling strategy those variables are provided prognostically by an ABL1d model. As far as the boundary conditions at $z = z_{\mathrm{top}}$ are concerned, Dirichlet boundary conditions $\mathbf{u}_h(z_{\mathrm{top}}) = \mathbf{u}_{\mathrm{LS}}(z_{\mathrm{top}})$ and $\phi(z_{\mathrm{top}}) = \phi_{\mathrm{LS}}(z_{\mathrm{
[revised manuscript text omitted]

 $l_{\text{D80}} = \sqrt{2e(z)/N^2}$. $l_{\text{up}}$ and $l_{\text{dwn}}$ are first initialized to $l_{\text{up}} = l_{\text{dwn}} = l_{\text{D80}}$. The resulting length scales are then limited  not only by the distance to the surface and to the top but also by the distance to a strongly stratified portion of the air column. This limitation amounts to control the vertical gradients of $l_{\text{up}}(z)$ and $l_{\text{dwn}}(z)$ such that they are not larger that the variations of

$$(l_{\text{up}})_{k-1/2} \equiv \min\left((l_{\text{up}})_{k+1/2} + \underline{\text{e3t}}_k, (l_{\text{D80}})_{k-1/2}\right)$$

$$(l_{\text{dwn}})_{k+1/2} \equiv \min\left((l_{\text{dwn}})_{k-1/2} + \underline{\text{e3t}}_k, (l_{\text{D80}})_{k+1/2}\right)$$

 altitude. The resulting mixing length will be simply referred to as $l_{\text{D80}}$. Note that the Taylor expansion of the integral in  (13) is

$$\int_z^{z+l_{\text{up}}} N^2(s)(s-z)ds \approx \frac{N^2(z)l_{\text{up}}^2}{2} + \frac{\frac{dN^2}{dz}l_{\text{up}}^3}{3} + \mathcal{O}(l_{\text{up}}^4),$$

which shows that the $l_{\text{D80}}$ mixing length is an approximation of $l_{\text{BL89}}$ which is obtained by retaining only the leading order term in the Taylor expansion.

**3.2.3**

3. **Rodier et al. (2017) length scale.** Recently, Rodier et al. (2017) proposed a modification of the Bougeault and Lacarrère (1989) mixing length. This modification turns out to improve results for stably stratified boundary layers typical of areas covered by ice. They propose to add a shear related term to (13) such that the definition of $l_{\text{up}}$ and $l_{\text{dwn}}$ becomes

$$\int_z^{z+l_{\text{up}}} \left[N^2(s)(s-z) + c_0\sqrt{e(s)}\|\partial_s \mathbf{u}_h\|\right] ds = e(z), \qquad \int_{z-l_{\text{dwn}}}^{z} \left[N^2(s)(z-s) + c_0\sqrt{e(s)}\|\partial_s \mathbf{u}_h\|\right] ds = e(z). \qquad (14)$$

where $c_0$ is a parameter whose value should be smaller than $\sqrt{C_m/c_\varepsilon}$. The value of $c_0$ will be chosen based on numerical experiments presented in Sec. 4.

$$\underline{\text{FC}(z_{p+1/2}) = \text{FC}(z_{p-1/2})} \quad \pm \quad \frac{\text{e3t}(z_p)}{2}\left(N^2(z_{p+1/2})(z_{p+1/2}-z_{k+1/2}) + N^2(z_{p-1/2})(z_{p-1/2}-z_{k+1/2})\right)$$

$$\pm \quad c_0\frac{\text{e3t}(z_p)}{2}\left(\sqrt{e(z_{p+1/2})}\|\partial_z\mathbf{u}_h(z_{p+1/2})\| + \sqrt{e(z_{p-1/2})}\|\partial_z\mathbf{u}_h(z_{p-1/2})\|\right)$$

In the following this mixing length will be referred to as $l_{\text{R17}}$.

**3.2.3**

4. **A local buoyancy and shear-based length scale.** For the sake of computational efficiency, we have derived a local version of the Rodier et al. (2017) length scale which is original to the present paper. Under the assumption that $l_{\text{up}}$ (resp. $l_{\text{dwn}}$) is small compared to the spatial variations of $N^2$, $e$, and $\|\partial_z\mathbf{u}_h\|$, we end up with the following second-order equation for $l_{\text{up}}$

$$\underline{\frac{N^2(z)}{2}l_{\text{up}}^2 + c_0\sqrt{e(z)}\|\partial_z\mathbf{u}_h\|l_{\text{up}} = e(z)}$$

$:\frac{N^2(z)}{2}l_{\text{up}}^2 + c_0\sqrt{e(z)}\|\partial_z\mathbf{u}_h\|l_{\text{up}} = e(z),$ whose unique positive solution is

$$l_{\text{D80}}^\star(z) = \frac{2\sqrt{e(z)}}{c_0\|\partial_z\mathbf{u}_h\| + \sqrt{c_0^2\|\partial_z\mathbf{u}_h\|^2 + 2N^2(z)}}.$$

We easily find that $l_{\text{D80}}^\star = l_{\text{D80}}$ for $\|\partial_z\mathbf{u}_h\| = 0$, and $l_{\text{D80}}^\star = \dfrac{\sqrt{e(z)}}{c_0\|\partial_z\mathbf{u}_h\|}$ for $N^2 = 0$ which is consistent with the shear based length scale of Wilson and Venayagamoorthy (2015). Once $l_{\text{D80}}^\star$ has been computed we apply the  as in the  $l_{\text{D80}}$ case.

The performance of those four length scales for various physical flows is discussed in  Sect. 4.

**3.3   Coupling with ocean and sea-ice**

For the practical implementation of the ABL coupling strategy within a global oceanic model, a proper coupling method is required for stability and consistency purposes  (e.g. Beljaars et al., 201 and the ABL1d must have the ability to handle grid cells partially covered by sea-ice. For the coupling strategy, a so-called implicit flux coupling which is unconditionally stable (App. B in Beljaars et al., 2017) and asymptotically consistent for $\Delta t \to 0$ (Renault et al., 2019a) is used. Because vertical diffusion in ABL1d is handled implicitly in time, the boundary conditions (3) should be provided at time $n+1$. The implicit flux coupling amounts to discretize the boundary conditions (3) as

$$K_m\partial_z\mathbf{u}_h|_{z=z_{\text{sfc}}}^{n+1} = C_D\|\mathbf{u}_h^n(z_{\text{sfc}}) - \widetilde{\mathbf{u}}_{\text{oce}}\|(\mathbf{u}_h^{n+1}(z_{\text{sfc}}) - \widetilde{\mathbf{u}}_{\text{oce}}) \tag{15}$$

$$K_s\partial_z\phi|_{z=z_{\text{sfc}}}^{n+1} = C_\phi\|\mathbf{u}_h^n(z_{\text{sfc}}) - \widetilde{\mathbf{u}}_{\text{oce}}\|(\phi^{n+1}(z_{\text{sfc}}) - \widetilde{\phi}_{\text{oce}}) \tag{16}$$

where $\widetilde{\mathbf{u}}_{\mathrm{oce}}$ and $\widetilde{\phi}_{\mathrm{oce}}$ are either the instantaneous values at time $n$ if NEMO and ABL1d have the same time-step or an average over the successive oceanic substeps otherwise.

A particular care has also been given to the compatibility between the ABL1d model and SI$^3$ (Sea Ice model Integrated Initiative) the sea-ice component of NEMO. SI$^3$ is a multi-category model whose state variables relevant for our study are the ice surface temperature $T_l^{\mathrm{ice}}$ with associated fractional area $a_l$ (for the $l^{\mathrm{th}}$ category), and the ice velocity $\mathbf{u}^{\mathrm{ice}}$ (same for all categories). Note that the values of the exchange coefficients over sea-ice $C_D^{\mathrm{ice}}$, $C_\theta^{\mathrm{ice}}$, and $C_q^{\mathrm{ice}}$ are different from their oceanic counterparts but are the same over all sea-ice categories. At this point there are several strategies for the ABL1d/SI$^3$ coupling:

1. run the ABL1d model over  the whole ABL for each category $l$ and then average atmospheric variables weighted by $a_l$,

2. run a single ABL1d model with a category averaged surface flux. In the current version of NEMO $C_\theta^{\mathrm{ice}}$ is function of the averaged temperature $T^{\mathrm{ice}}$ which means that it is equivalent to compute a flux over each category before averaging them and to compute a single flux using the averaged surface temperature, indeed

$$\sum_l a_l \left[ C_\theta^{\mathrm{ice}} \| \mathbf{u}_h(z_{\mathrm{sfc}}) - \mathbf{u}^{\mathrm{ice}} \| (\theta(z_{\mathrm{sfc}}) - T_l^{\mathrm{ice}}) \right] = C_\theta^{\mathrm{ice}} \| \mathbf{u}_h(z_{\mathrm{sfc}}) - \mathbf{u}^{\mathrm{ice}} \| \left( \theta(z_{\mathrm{sfc}}) - \sum_l a_l T_l^{\mathrm{ice}} \right)$$

The second option has been preferred because it is much easier to implement and more computationally efficient. It amounts to consider an ice surface temperature averaged over all categories $T^{\mathrm{ice}} = \sum_{l=1}^{n_{\mathrm{cat}}} a_l T_l^{\mathrm{ice}}$ for the computation of ice-atmosphere turbulent fluxes ($T^{\mathrm{ice}}$ also enters in the computation of $q_{\mathrm{ice}}$). Noting $F_{\mathrm{oce}}$ the fraction of open water (lead), the boundary condition (15) and (16) are modified in

$$K_m \partial_z \mathbf{u}_h |_{z=z_{\mathrm{sfc}}}^{n+1} = F_{\mathrm{oce}} C_D \| \mathbf{u}_h^n(z_{\mathrm{sfc}}) - \widetilde{\mathbf{u}}_{\mathrm{oce}} \| (\mathbf{u}_h^{n+1}(z_{\mathrm{sfc}}) - \widetilde{\mathbf{u}}_{\mathrm{oce}}) + (1 - F_{\mathrm{oce}}) C_D^{\mathrm{ice}} \| \mathbf{u}_h^n(z_{\mathrm{sfc}}) - \widetilde{\mathbf{u}}^{\mathrm{ice}} \| (\mathbf{u}_h^{n+1}(z_{\mathrm{sfc}}) - \widetilde{\mathbf{u}}^{\mathrm{ice}})$$

$$K_s \partial_z \phi |_{z=z_{\mathrm{sfc}}}^{n+1} = F_{\mathrm{oce}} C_\phi \| \mathbf{u}_h^n(z_{\mathrm{sfc}}) - \widetilde{\mathbf{u}}_{\mathrm{oce}} \| (\phi^{n+1}(z_{\mathrm{sfc}}) - \widetilde{\phi}_{\mathrm{oce}}) + (1 - F_{\mathrm{oce}}) C_\phi^{\mathrm{ice}} \| \mathbf{u}_h^n(z_{\mathrm{sfc}}) - \widetilde{\mathbf{u}}^{\mathrm{ice}} \| (\phi^{n+1}(z_{\mathrm{sfc}}) - \widetilde{\phi}^{\mathrm{
[revised manuscript text omitted]